# PD-1⁻ CD45RA⁺ effector-memory CD8 T cells and CXCL10⁺ macrophages are associated with response to atezolizumab plus bevacizumab in advanced hepatocellular carcinoma

Sarah Cappuyns [1,2,3,4], Gino Philips[3,4], Vincent Vandecaveye[5,6], Bram Boeckx[3,4], Rogier Schepers[3,4], Thomas Van Brussel[3,4], Ingrid Arijs [3,4], Aurelie Mechels [3,4], Ayse Bassez[3,4], Francesca Lodi[3,4], Joris Jaekers[7], Halit Topal[7], Baki Topal[7], Orian Bricard[3,4], Junbin Qian [3,4,8,9], Eric Van Cutsem[1,2], Chris Verslype[1,2], Diether Lambrechts [3,4,10] ✉ & Jeroen Dekervel [1,2,10] ✉

The combination of atezolizumab plus bevacizumab (atezo/bev) has dramatically changed the treatment landscape of advanced HCC (aHCC), achieving durable responses in some patients. Using single-cell transcriptomics, we characterize the intra-tumoural and peripheral immune context of patients with aHCC treated with atezo/bev. Tumours from patients with durable responses are enriched for PDL1⁺ CXCL10⁺ macrophages and, based on cell−cell interaction analysis, express high levels of *CXCL9/10/11* and are predicted to attract peripheral *CXCR3*⁺ CD8⁺ effector-memory T cells (CD8 T$_{EM}$) into the tumour. Based on T cell receptor sharing and pseudotime trajectory analysis, we propose that CD8 T$_{EM}$ preferentially differentiate into clonally-expanded PD1⁻ CD45RA⁺ effector-memory CD8⁺ T cells (CD8 T$_{EMRA}$) with pronounced cytotoxicity. In contrast, in non-responders, CD8 T$_{EM}$ remain frozen in their effector-memory state. Finally, in responders, CD8 T$_{EMRA}$ display a high degree of T cell receptor sharing with blood, consistent with their patrolling activity. These findings may help understand the possible mechanisms underlying response to atezo/bev in aHCC.

Hepatocellular carcinoma (HCC) is the most common form of liver cancer and one of the few neoplasms with increasing incidence and mortality worldwide[1]. The majority of HCC patients (50-60%) eventually evolve to an advanced stage (aHCC) requiring systemic treatment[2]. Like in other cancer types, immune checkpoint inhibitors (CPI) have dramatically changed the treatment landscape of aHCC. In front line clinical trials, the combination of the PDL1 inhibitor, atezolizumab, with the anti-VEGFA antibody, bevacizumab, demonstrated

median overall survival (OS) of 19.2 months[3,4], which is almost double compared to results achieved using tyrosine kinase inhibitors (TKI)[5,6] or anti-PD(L)1 monotherapy[7–11]. In the adjuvant setting, dual PDL1/ VEGFA inhibition has also shown promise to reduce the risk of recurrence after curative resection or ablation[12].

Despite great efforts to characterize the tumour-microenvironment (TME) of HCC[13–17], factors associated with response/resistance to the combination of atezolizumab plus

bevacizumab (atezo/bev) remain to be elucidated. PD1-expressing CD8 T cells have been identified as key effector cells in response to PD(L)1 inhibition in several tumour types, including breast cancer[18], lung cancer[19] and melanoma[20], where persistent exposure of CD8 T cells to tumour antigens will stimulate differentiation towards a dysfunctional, exhausted phenotype. CD8 T cells express typical exhaustion markers, such as *PDCD1* (PD1), upon which their activation status and anti-tumoural cytolytic function are dampened. PD(L)1 blockade reinvigorates the anti-tumoural immune response leading to proliferation of these cytotoxic T cells that are able to overcome tumour-induced immunosuppression and induce durable clinical benefits. However, in HCC, the role of PD1$^+$ CD8 T cells is controversial. The presence of exhausted PD1$^+$ CD8 T cells in HCC has been associated with a more aggressive disease biology[21] and poor prognosis[22]. In pre-clinical models of non-alcoholic steatohepatitis (NASH)-associated HCC, exhausted, unconventionally activated PD1$^+$ CD8 T cells were related to impaired tumour surveillance, causing tissue damage and facilitating, rather than inhibiting, hepatocarcinogenesis upon anti-PD1 treatment[23]. Moreover, response to the combination of atezo/bev has been linked to a pre-existing immune response characterized by an increased CD8 T cell infiltration as well as overexpression of *CD274* (PDL1), and not *PDCD1* (PD1)[24]. The differences between HCC and other solid tumour types might be in part explained by the unique immune context of the liver as well as the fact that liver cancer most often develops in a background of chronic inflammation caused by a variety of chronic liver diseases[2].

Understanding which effector cells facilitate response to atezo/bev in aHCC and characterizing the optimal pre-treatment immune context for subsequent treatment success is crucial. It would not only enable the identification of patients likely to respond to atezo/bev, currently still an important unmet clinical need, but also contribute to the development of new strategies for the majority of aHCC patients who experience rapid disease progression and poor outcomes with current treatment options.

Here, we use pre-treatment tissue biopsies and serial peripheral blood mononuclear cell (PBMC) samples from patients with aHCC (n = 44) treated with systemic therapy to single-cell transcriptome (scRNAseq) and T cell receptor sequencing (scTCRseq). Patients treated with atezo/bev (n = 25) were stratified according to clinical response and various single-cell readouts were correlated with response and clinical outcome. We report that, in tumours responding to atezo/bev, CXCR3$^+$ effector memory T cells differentiated primarily towards PD1$^-$ CD45RA$^+$ effector-memory CD8 T cells. This population was clonally expanded and characterized by a high degree of TCR sharing with peripheral blood. In addition, the intra-tumoural presence of this population prior to treatment is associated with durable response to atezo/bev in aHCC. Furthermore, PDL1$^+$ CXCL10$^+$ macrophages are enriched in responding tumours, where they interact with the peripheral T cell compartment ensuring effective recruitment of primed effector-memory T cells into the TME. These findings were validated in transcriptomic data of aHCC patients treated with atezo/bev versus sorafenib[3,4], confirming CD45RA effector-memory CD8 T cells and CXCL10$^+$ macrophages as potential predictive biomarkers of response to atezo/bev in aHCC.

## Results

### The tumour microenvironment and peripheral immune system of advanced HCC

For 38 (out of 44) aHCC patients, a pre-treatment tissue biopsy was subjected to scRNAseq (Fig. 1a; Supplementary Table 1 and 2), yielding high quality transcriptomic data from 97 947 single-cells (Fig. 1b). Subsequent analysis involving dimensionality reduction and clustering identified several clusters, assigned to T cells and NK-cells (30%), B-cells (5%), myeloid cells (12%) and stromal cell types (12%) based on marker gene expression (Supplementary Fig. 1a). We also identified a

proliferative cluster, which mainly consisted of proliferating T cells (Supplementary Fig. 1b) and a large cluster of HCC cancer cells (40%) expressing both genes associated with normal liver function (*ALB, HP, FGA, FGB*) and liver cancer (*AFP, SPINK1, GPC3, AKR1C1*). Inferring copy number variations (CNV) from the scRNAseq data[25] confirmed the malignant origin of the HCC cluster that displayed CNV alterations previously described in HCC (Supplementary Fig. 1c)[26]. There was no evidence of cluster bias based on underlying liver disease, treatment or biopsy type in immune cells and stromal cell types (Supplementary Fig. 1d). Similarly, single-cell profiling of serial on-treatment (week 0-3-6) PBMC samples (n = 72 from 25 aHCC patients; Supplementary Table 2), yielded high-quality transcriptomic data for 268 807 PBMCs, annotated to their respective cell types using marker genes (Fig. 1c; Supplementary Fig. 1e). Together, these data provide a unique cell atlas of both the TME and peripheral immune system of aHCC patients treated with systemic therapies and an invaluable resource for future research endeavours (see Data Availability).

### The intra-tumoural T-/NK-cell composition is distinct from the peripheral T-/NK-cell composition

First, we explored the T-/NK-cell compartment of the TME and peripheral blood in more detail. Sub-clustering a total of 26 380 intra-tumoural T-/NK-cells and 170 919 peripheral T-/NK-cells (64% of PBMCs) separately, we identified several phenotypes of CD4 T cells, CD8 T cells and natural killer cells (NK cells) (Fig. 2a, b; Supplementary Fig. 2a, b). Notably, CD4 (CD4 CXCL13) and CD8 (CD8 T$_{EX}$) 'exhausted' T cells were unique to the TME and characterized by the highest expression of *PDCD1* (PD1) and other known exhaustion markers (Supplementary Fig. 2c–e). Importantly, CD8 T$_{EX}$ expressed the highest levels of *IFNG* along with a number of cytotoxic markers (*PRF1, NKG7*; Supplementary Fig. 2e), despite their 'exhaustion' phenotype, supporting their denomination as 'antigen-experienced' T cells[18]. On the other hand, CD4 CXCL13 have been previously described as 'exhausted' CD4 T cells in HCC[13,14]. Though this cluster was very small in aHCC, previous studies[18] suggest that it consists of both Th1 CD4 T cells (*IFNG, CXCR3*) and follicular-helper CD4 T cells (*BCL6, CD200*)[27]. Finally, based on the expression of typical marker genes (*CX3CR1, SPON2, FGFBP2*; Supplementary Fig. 2a), we identified CD45RA effector-memory CD8 T cells (CD8 T$_{EMRA}$) both in the TME and in peripheral blood. We confirmed their expression of CD45RA at the protein level using TotalSeq-C data (Supplementary Fig. 2f). Importantly, CD8 T$_{EMRA}$ were phenotypically similar and clustering close to the cytotoxic NK-cells, but distinguishable based on their expression of CD8 (*CD8A, CD8B*; Supplementary Fig. 2g) and the detection of a productive TCR sequence (Supplementary Fig. 2h).

Combining scRNAseq and scTCRseq, we identified 17 842 T cells carrying 12 690 unique TCRs in the TME, while 115 711 peripheral T cells carried 90 188 unique TCR sequences. We identified TCR clonotypes based on identical TCR sequences, and defined dominant clonotypes as TCRs shared by >5 T cells. In intra-tumoural T cells, dominant clonotypes were concentrated within effector (CD8 T$_{EM}$, CD8 T$_{EMRA}$) and 'antigen-experienced' T cell clusters (CD8 T$_{EX}$, CD4 CXCL13; Fig. 2c), while non-dominant clonotypes were mostly found in naive, memory or regulatory T cell subtypes. Similarly, dominant peripheral T cell clonotypes, in line with their phenotypical counterparts in the tumour, were concentrated within peripheral effector T cells (CD8 T$_{EM}$, CD8 T$_{EMRA}$, CD4 T$_{CYTO}$; Fig. 2d). These findings were replicated when defining dominant clonotypes as TCRs representing 1% or more of the TCR repertoire (Supplementary Fig. 3a, b) in order to account for the number of T cells detected.

In short, though phenotypically distinct, both CD8 effector T cells (CD8 T$_{EM}$, CD8 T$_{EMRA}$) and CD8 exhausted T cells (CD8 T$_{EX}$) were characterized by dominant T cell clonotypes both in the tumour and in peripheral blood.

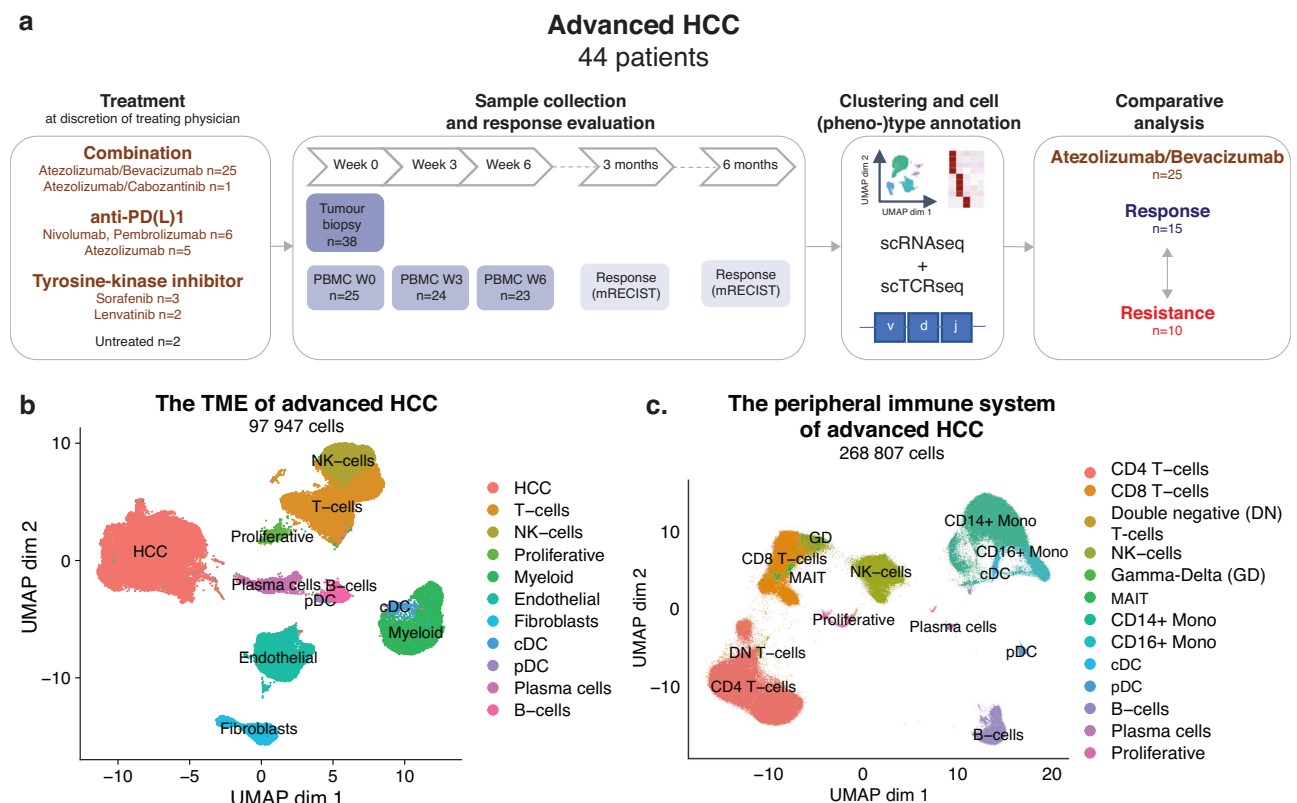

**Fig. 1 | The tumour-microenvironment and peripheral immune system of advanced HCC. a**. Study design, patient and sample overview. **b**. UMAP depicting cell types identified in pre-treatment TME of advanced HCC (n = 38 tumour biopsies). **c**. UMAP representation of cell types identified in blood (n = 72 PBMC samples). (cDC, conventional dendritic cells; pDC, plasmacytoid dendritic cells; DN T cells, Double negative T cells; GD T cells, Gamma-delta T cells; HCC, hepatocellular carcinoma; PD1, Programmed cell death protein 1; PDL1, Programmed death-ligand 1; scRNAseq, single-cell RNA sequencing; scTCRseq, single-cell T cell receptor sequencing; TME, tumour-microenvironment; UMAP, Uniform Manifold Approximation and Projection).

## Clonally-expanded CD8 T$_{EMRA}$ are associated with response to atezolizumab/bevacizumab

In order to identify those CD8 T cells phenotypes associated with response to atezo/bev, we compared several characteristics of the tumoural and peripheral immune system between responders and non-responders, including i) abundancies of intra-tumoural CD8 T cells, ii) the tumoural and peripheral TCR repertoire and iii) the degree of TCR sharing between tumour and blood.

First, comparing relative abundancies of various T cell phenotypes in the TME between responders and non-responders, we found that CD8 T$_{EMRA}$ were more abundant in responding tumours (p = 0.04), while CD8 T$_{EM}$ were increased in non-responders (p = 0.005; Fig. 2e; Supplementary Fig. 3c). Of note, though CD8 T$_{EX}$ expressed the highest levels of the therapeutic target *PDCD1* (PD1; Supplementary Fig. 3d, e), their presence (*i.e.* relative abundancies) in the TME did not differ according to response (Fig. 2e; Supplementary Fig. 3c). Differential gene expression of CD8 T cells in the TME revealed upregulation of cytotoxic genes (*GZMB, GNLY, PRF1, GZMH)* and typical CD8 T$_{EMRA}$ markers (*FGFBP2, FCGR3A*), suggesting that CD8 T$_{EMRA}$ might play an important role in achieving durable response to atezo/bev (Fig. 2f). In contrast, non-responding tumours were more memory-like (*FOS)* and, upregulated *GZMK*, a typical CD8 T$_{EM}$ marker (Fig. 2f). Secondly, responding tumours were characterized by a more clonal pre-treatment TCR repertoire while non-responders displayed a richer and more diverse, non-clonal baseline TCR repertoire (Fig. 2g; Supplementary Fig. 3f). Calculating the Gini-index, which takes both TCR evenness (1-clonality) and TCR richness into account, for each CD8 T cell phenotype, intra-tumoural CD8 T$_{EM}$, CD8 T$_{EMRA}$ and CD8 T$_{EX}$ were most clonally-expanded (Fig. 2h *left*). Importantly, when stratifying for

response to atezo/bev, CD8 T$_{EMRA}$ in responding tumours had a significantly higher Gini-index compared to non-responders (p = 0.045; Fig. 2h *right*). In contrast, we did not detect significant differences in clonal expansion of CD8 T$_{EM}$ and CD8 T$_{EX}$ when comparing responders to non-responders. These findings suggest that clonally expanded CD8 T$_{EMRA}$ residing within the TME prior to treatment may facilitate response to atezo/bev in aHCC.

## TCR sharing confirms CD8 T$_{EMRA}$ as crucial effector T cells in the TME of aHCC

As intra-tumoural T cells that share identical TCR sequences with T cells residing in peripheral blood[28,29] are more likely to be tumour-reactive, we explored TCRs shared between tumours and PBMCs in 17 atezo/bev-treated patients, 10 of which were atezo/bev-responders. We focussed specifically on those TCR sequences present in tumour and peripheral blood prior to treatment initiation (PBMC week 0), hypothesizing that these shared TCRs represent a baseline immune response, directed at and driven by the tumour. A total of 403 unique shared, potentially 'tumour-specific', TCRs were detected, representing approximately 7.1% of all TCRs detected in the tumour compared to 0.6% of all TCRs detected in peripheral blood. In order to correct for the number of T cells detected in each sample, we calculated the proportion of shared TCRs relative to the total number of TCRs detected in PBMCs and found that responders displayed a higher degree of TCR sharing (on average 2%; p = 0.03; Fig. 3a). Similar trends were detected in proportions of shared peripheral T cells (Supplementary Fig. 3g). Importantly, increased TCR sharing was associated with significantly longer PFS (median PFS 12 versus 2 months; p = 0.012; Fig. 3b), supporting our hypothesis that TCR sharing may

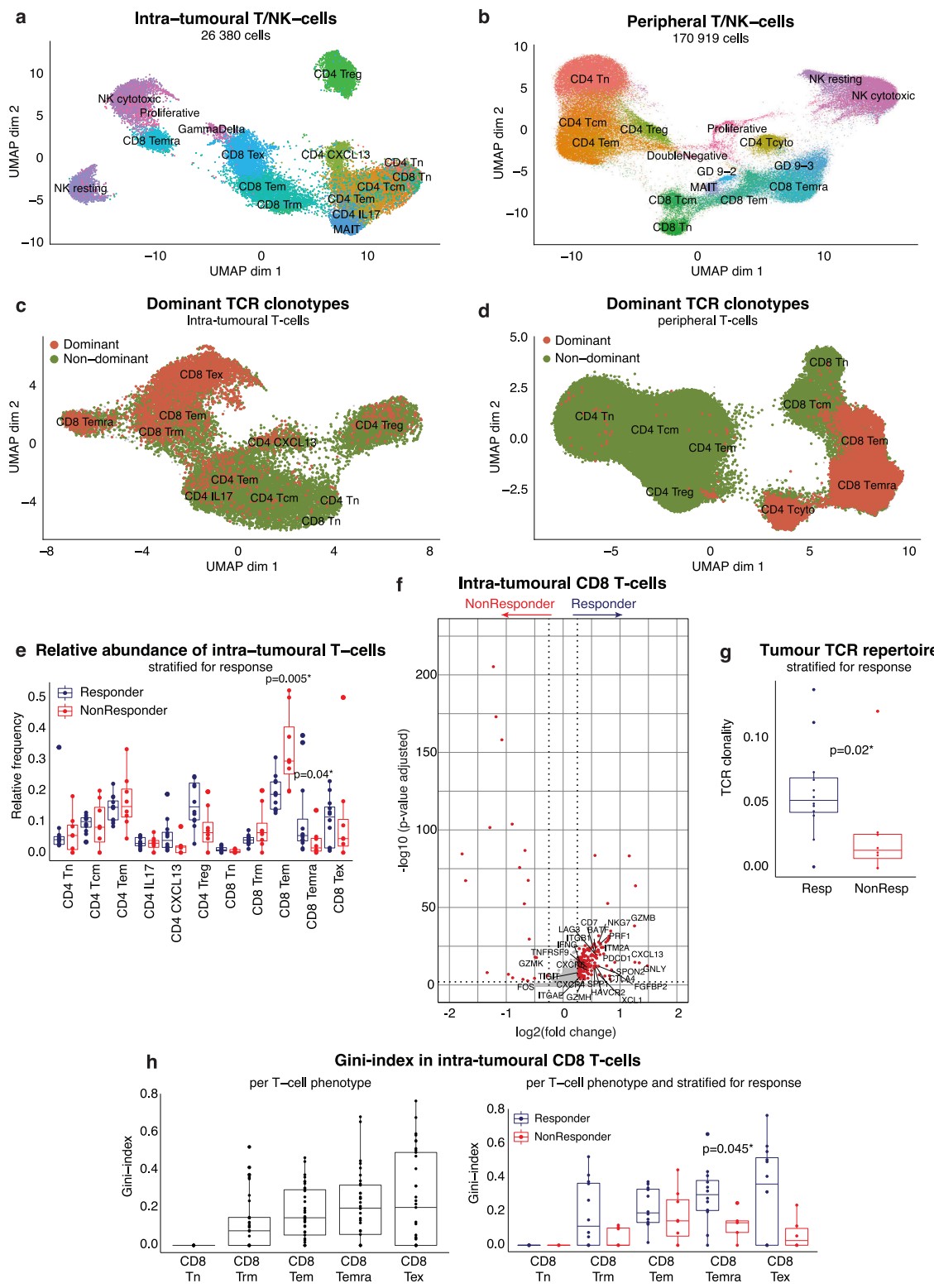

indeed identify the fraction of intra-tumoural T cells that truly target the tumour.

Linking these shared TCRs to their T cell phenotype in the TME (Fig. 3c *left*) revealed that the majority represented CD8 T cells, concentrated within CD8 effector subtypes (CD8 $T_{EM}$ and CD8 $T_{EMRA}$). In fact, 63% of CD8 $T_{EMRA}$ and 22% of CD8 $T_{EM}$ in the TME were characterized by a TCR also detected in peripheral blood prior to treatment, while CD8 $T_{EX}$ displayed far less TCR sharing with peripheral blood (13%; Supplementary Fig. 3h). In line with these findings, differential gene expression, demonstrated an overexpression of CD8 (*CD8A* and *CD8B*) and cytotoxic markers (*GZMA, GZMB, GZMH, GNLY, PRF1*), as well as typical CD8 $T_{EMRA}$ markers (*CX3CR1, SPON2, FGFBP2)* in intra-tumoural T cells with shared TCRs (n = 970 shared T cells). In contrast, T cells carrying a TCR found exclusively in the tumour were enriched for exhaustion markers (*CTLA4*) and regulatory genes (*FOXP3, TNFRSF4*; Fig. 3d). Importantly, while shared CD8 $T_{EM}$ were present in the TME of both responders and non-responders (18% versus 8% of all intra-tumoural CD8 $T_{EM}$ respectively), shared CD8 $T_{EMRA}$,

**Fig. 2 | Clonally-expanded CD8 T_EMRA are associated with response to atezolizumab/bevacizumab. a.** UMAP representation depicting intra-tumoural T/NK-cell phenotypes (n = 38 tumour biopsies). **b.** UMAP representation of peripheral T/NK-cell phenotypes(n = 72 PBMC samples). **c.** UMAP representation of dominant versus non-dominant clonotypes in intra-tumoural T cells (n = 38 tumour biopsies). Dominant clonotypes were defined as TCR sequences shared by >5 T cells. **d.** UMAP representation of dominant versus non-dominant clonotypes in peripheral T cells (n = 72 PBMC samples). Dominant clonotypes were defined as TCR sequences shared by >5 T cells. **e.** Boxplots depicting relative abundance of intra-tumoural T-/NK-cell phenotypes in atezo/bev-treated patients (n = 20), calculated per patient and stratified for response (12 Resp versus 8 NonResp). P-values calculated using two-sided Mann-Whitney U-test, only p-values < 0.05 are shown. Boxes indicate median +/- interquartile range; whiskers show minima and maxima. **f.** Volcano plot depicting differentially expressed genes in intra-tumoural CD8 T cells (n = 4313) from responders (n = 12; 3425 CD8 T cells) versus non-responders (n = 8; 888 CD8 T cells). P-values were obtained using the two-sided Wilcoxon test and Bonferroni-corrected (Seurat 4[53]). *Red*: adjusted p-value < 0.01 and log$_2$ fold change >0.25. **g.** TCR clonality of intra-tumoural T cells in atezo/bev-treated patients (n = 20), calculated per patient and stratified for response (12 Resp versus 8 NonResp). P-value calculated using Mann-Whitney U-test. Boxes indicate median +/- interquartile range; whiskers show minima and maxima. **h.** Gini-index of intra-tumoural CD8 T cells. *Left*: calculated per patient (n = 37), per CD8 T cell phenotype. *Right:* in atezo/bev-treated patients (n = 20), calculated per patient and stratified for response (12 Resp versus 8 NonResp). P-values calculated using two-sided Mann-Whitney U-test, only p-values < 0.05 are shown. Boxes indicate median +/- inter-quartile range; whiskers show minima and maxima. (GD, γδ T cells; TCR, T cell receptor; UMAP, Uniform Manifold Approximation and Projection; Resp, responder; NonResp, non-responder).

were almost exclusively seen in responding tumours (57% of all intra-tumoural CD8 T_EMRA in responders, compared to 6.1% in non-responders; Fig. 3c *right*).

CD8 T_EMRA have been described as 'recently-activated' CD8 effector-memory T cells[30]. They do not express *PDCD1* or other markers traditionally associated with antigen-experience. Instead they re-express *CD45RA* after antigenic stimulation[31,32]. They are considered a sentinel-like T cell phenotype that patrols inflammatory sites of frequent antigenic encounter[33]. They are endowed with potent cytolytic properties based on their high expression of cytotoxic markers (*PRF1, NKG7, GZMA, GZMB, GZMH, GNLY*; Fig. 3e) that relies on direct interaction between the T cell and its target cell and constitutively express receptors that direct their migration to inflamed tissue (*CX3CR1*)[33].

### Interaction with tumour-antigens drives intra-tumoural differentiation towards CD8 T_EMRA

In order to gain insights into the origins of CD8 T_EMRA in the TME, we computed pseudotime trajectories of intra-tumoural CD8 T cells using Slingshot[34]. We considered CD8 naive T cells (CD8 T_N) as the root of the trajectory because they had the highest TCR richness (Supplementary Fig. 4a). In line with previous reports[13,18], naive T cells were connected to T_EM cells and then diverged into three distinct trajectories, connecting naive and T_EM T cells to T_RM, T_EMRA and T_EX (Fig. 4a *left*). TCR richness decreased along each of these trajectories (Supplementary Fig. 4b). CD8 T_EM displayed most TCR clonotype overlap with T_EX, but also with T_EMRA and T_RM (Fig. 4b), while there was almost no TCR overlap between T cells belonging to different lineages, supporting the validity of the three trajectories. Profiling marker genes along each trajectory confirmed their functional annotation (Supplementary Fig. 4c).

When plotting the densities of T cells along each trajectory, we found striking differences between responders and non-responders. Intra-tumoural CD8 T cells in responders were capable of evolving towards more differentiated phenotypes, an effect that was most pronounced in the T_EMRA trajectory, while non-responders seemed frozen at an earlier stage of the pseudotime (p < 0.001; Fig. 4c). There was a steady increase in Gini-index along both the T_EMRA and T_EX trajectories of responding tumours. Importantly, when assessing the density of shared T cells along the CD8 trajectories (Fig. 4a *right*), we found these were clearly enriched towards the end of the T_EMRA trajectory in responders (Fig. 4d). In contrast, along the T_EX trajectory, the greatest T cell density was seen at the T_EM stage in both responders and non-responders.

We then used TradeSeq[35] to identify sets of genes differentially expressed along the T_EMRA versus T_EX trajectories (using *diffEnd* test). A total of 13 pathways and 46 pathways from the REACTOME or the 'GO: biological processes' gene sets were significantly enriched in the CD8 T_EMRA and T_EX trajectories, respectively. Importantly, the T_EMRA trajectory was dominated by pathways related to innate-like immunity (Fig. 4e), reflecting their role as potent effector T cells that eliminate cancer cells through direct cytotoxicity. In contrast, the T_EX trajectory was enriched in pathways involved in *IFNG* signalling and immune cell activation and differentiation. In order to understand which factors drive this dual differentiation, we again used TradeSeq[35] to assess differences in expression patterns before and after the point where the trajectories diverge (using *earlyDEG* test) and found a total of 333 pathways to be enriched. Importantly, pathways involved in antigen-binding were top ranked, suggesting that further differentiation requires direct interaction with antigens (Fig. 4f).

Finally, to study the on-treatment immune response, we used shared TCR clonotypes present in PBMCs and the TME prior to treatment, linked them to their phenotype in peripheral blood and tracked their evolution during treatment in PBMCs sampled during treatment (week 0-3-6). Firstly, the 422 unique TCRs characterizing CD8 T_EMRA in the TME were found predominantly in peripheral CD8 T_EMRA. Prior to treatment they represented 18% of all CD8 peripheral T cells in responders (1183 out of 6575 peripheral CD8 T cells), compared to just 2% in non-responders (Fig. 4g *left*). Tracking their evolution during treatment, the degree of TCR sharing remained high in responders (18.6% after 6 weeks), and we did not observe any significant changes in non-responders (1.6% at week 6; Fig. 4g *left*). This was in stark contrast to the 1065 unique TCRs found in CD8 T_EX in the TME that were found in less than 1% of CD8 peripheral T cells in responders and non-responders alike (Fig. 4g *right*). Moreover, these TCRs found in intra-tumoural CD8 T_EX did not emerge in peripheral blood during treatment.

Taken together, these data suggest that while CD8 T_EM are present in the TME of both responders and non-responders alike, upon stimulation by tumoural antigens, CD8 T_EM are more likely to differentiate into CD8 T_EMRA in responders specifically, potentially resulting in direct anti-tumour cytotoxicity. Furthermore, CD8 T_EMRA display significant TCR sharing with PBMC in responders, and continue to do so upon treatment with atezo/bev, in line with their patrolling phenotype. In contrast, intra-tumoural differentiation from CD8 T_EM to CD8 T_EX occurs equally in responders and non-responders to atezo/bev and CD8 T_EX do not share TCRs with blood prior to treatment, nor do they appear during treatment with atezo/bev.

### Pro-inflammatory PDL1-expressing CXCL10+ macrophages are associated with response

While intra-tumoural CD8 T_EMRA were associated with subsequent response to atezo/bev, they do not express *PDCD1* (PD1). Therefore, we wondered whether the true target of atezo/bev in aHCC might be found in PDL1-expressing cells. Expression of *CD274* (PDL1) in the TME was generally low, but clearly detectable in myeloid cells (Supplementary Fig. 5a, b). Therefore, we subclustered the 11 678 myeloid cells into monocytes/macrophages (n = 10 609) and dendritic cells (DC; n = 764; Supplementary Fig. 5c, d). Within the monocyte/macrophage compartment, we identified several tumour-associated macrophage

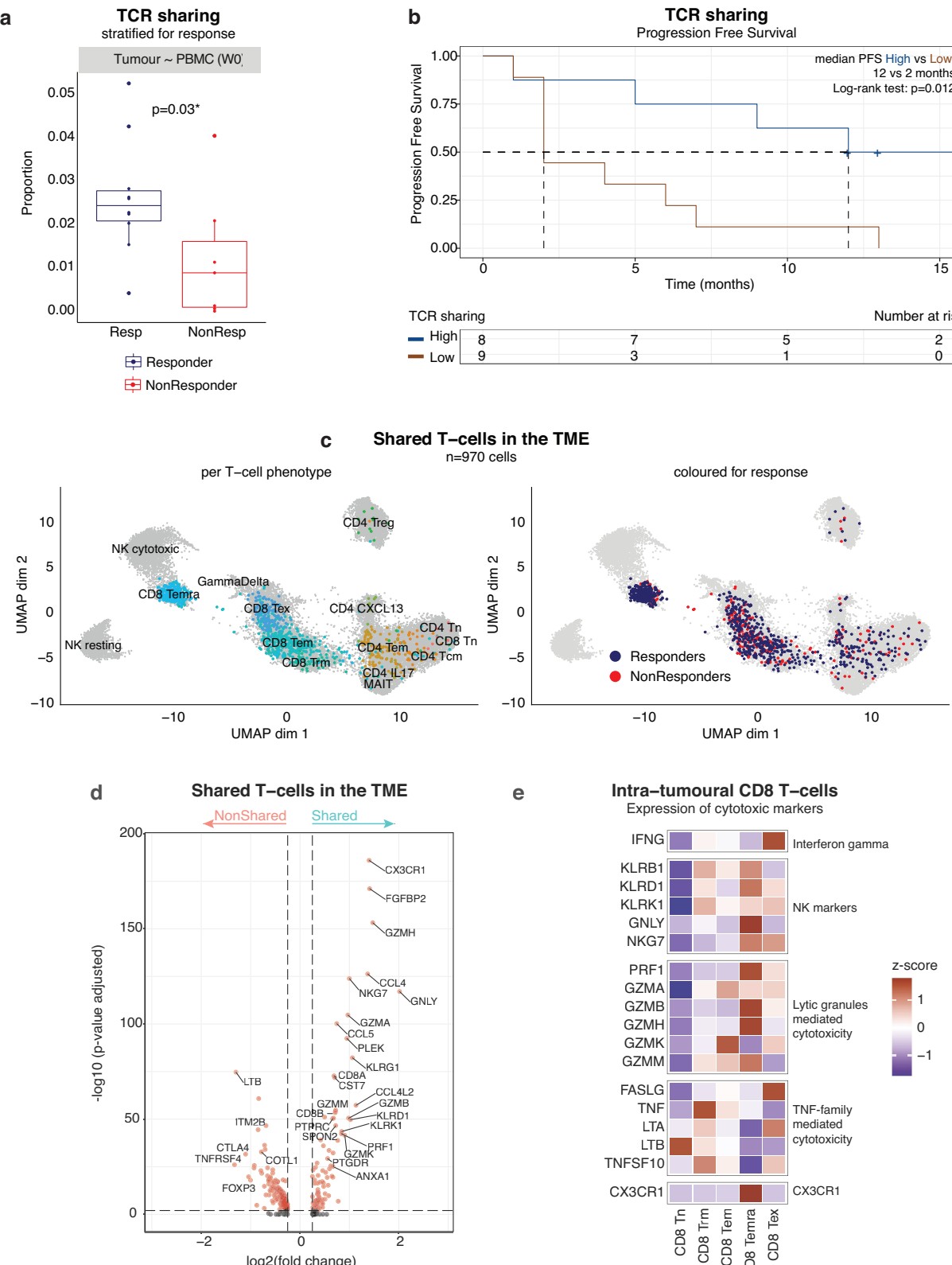

**a** TCR sharing — stratified for response — Tumour ~ PBMC (W0) — p=0.03*

**b** TCR sharing — Progression Free Survival — median PFS High vs Low: 12 vs 2 months — Log-rank test: p=0.012

| TCR sharing | Number at risk | | | |
|---|---|---|---|---|
| High | 8 | 7 | 5 | 2 |
| Low | 9 | 3 | 1 | 0 |

**c** Shared T−cells in the TME — n=970 cells — per T−cell phenotype — coloured for response

**d** Shared T−cells in the TME

**e** Intra−tumoural CD8 T−cells — Expression of cytotoxic markers

(TAM) subtypes (Fig. 5a; Supplementary Fig. 5e), of which the majority expressed high levels of anti-inflammatory markers, suggesting a predominantly immunosuppressive baseline TME in aHCC (Supplementary Fig. 5f). However, we also identified pro-inflammatory CXCL10⁺ TAMs (Macro CXCL10) characterized by high expression of genes involved in T cell recruitment (*CXCL9, CXCL10*) and interferon-gamma signalling (*STAT1, IDO1, GBP1*) (Supplementary Fig. 5f).

Comparing relative abundancies of the various TAM subtypes between responding and non-responding tumours, we found non-responders to have a higher abundance of TREM2-expressing macrophages, previously identified as immunosuppressive macrophages in HCC tumours[36], where their presence has been linked to resistance to anti-PD1 therapy[18,37,38] (Supplementary Fig. 6a). Differential gene expression showed an enrichment of genes involved in T cell

**Fig. 3 | TCR sharing confirms CD8 T$_{EMRA}$ as crucial effector T cells in the TME of aHCC. a**. Proportion of TCRs shared between tumour and blood prior to treatment (PBMC week 0), relative to the total number of TCRs detected, calculated per sample (n = 17) and stratified for response (10 Resp versus 7 NonResp). P-values calculated using two-sided Mann-Whitney U-test. Boxes indicate median +/- interquartile range; whiskers show minima and maxima. **b**. Kaplan-Meier plot of progression free survival in atezo/bev-treated patients (n = 17; 10 Resp versus 7 NonResp) with high or low (split by median) TCR sharing between tumour and blood. **c**. UMAP representation of T cells characterized by a TCR shared between tumour and blood prior to treatment (n = 970 T cells). *Left:* coloured per T cell phenotype. *Right:* coloured for response to atezo/bev. **d**. Volcano plot depicting differentially expressed genes in shared (i.e. intra-tumoural T cells characterized by a TCR found in PBMC week 0; n = 970 T cells) versus non-shared T cells in the TME. P-values were obtained using the two-sided Wilcoxon test and Bonferroni-corrected (Seurat 4[53]). *Red*: adjusted p-value < 0.01 and log$_2$ fold change >0.25. **e**. Heatmaps showing expression of cytotoxic genes in intra-tumoural CD8 T cells. (PFS, Progression free survival; Resp, responder; NonResp, non-responder; TCR, T cell receptor; TME, tumour-microenvironment; UMAP, Uniform Manifold Approximation and Projection).

recruitment (*CXCL9, CXLC10*) and interferon-gamma activity (*GBP1, STAT1*) in the macrophage compartment of responding tumours, while non-responders were enriched in immunosuppressive markers (*GPNMB, CCL18;* Fig. 5b). Responding tumours also displayed higher levels of *CCL2*, which is a potent monocyte-attracting chemokine[39,40], but is also involved in recruitment of other immune cells into the TME. Finally, responders displayed higher expression of *SPP1*, previously associated with response to CPI monotherapy in lung cancer[41]. Additional pathway analysis confirmed that macrophages of responding tumours were enriched in pro-inflammatory pathways (Supplementary Fig. 6b). Importantly, on average, myeloid cells from responders expressed significantly higher levels of *CD274* (Fig. 5c *top*). More specifically, *CD274* expression was highest in Macro CXCL10 (Supplementary Fig. 6c, d) and Macro CXCL10 derived from responding tumours displayed higher *CD274* expression (Fig. 5c *bottom*). High *CD274* (PDL1) expression in Macro CXCL10 was also associated with longer PFS (median PFS 13 versus 3 months; p = 0.035; Fig. 5d). Taken together, response to atezo/bev is associated with an activated, pro-inflammatory, PDL1-expressing myeloid component in the pre-treatment TME.

### PDL1-expressing CXCL10$^+$ macrophages recruit effector-memory T cells into the TME

Tumour-associated macrophages have been associated with recruitment of peripheral T cells into the TME[42]. Therefore, we used CellChat[43] to predict receptor-ligand interactions between myeloid cells and T cells. Firstly, calculating the significant interactions between immune cell types in the TME separately for responders and non-responders, we found that overall, responders displayed more interaction possibilities (Supplementary Fig. 6e). Focussing on the CXCL signalling pathway network, we found predicted interactions between liver-resident macrophages (Kupffer cells) and T cells in all patients, regardless of response. In contrast, when compared to non-responders, responding tumours displayed more predicted interactions between CXCL10$^+$ macrophages and the T cell compartment (Fig. 5e). While the CXCL12/CXCR4 interaction, which originated almost exclusively from Kupffer cells, was found both in responders and non-responders, the *CXCL9/10/11* and *CXCR3* ligand-receptor pairs were significantly enriched in responders compared to non-responders (Fig. 5f).

In the TME, *CXCR3* was prominently expressed in several activated T cell subtypes (i.e. CD4 CXCL13 and CD8 T$_{EX}$, in addition to CD4 T$_{EM}$ and CD8 T$_{EM}$, Fig. 5g), but not in CD8 T$_{EMRA}$ and along the T$_{EMRA}$ trajectory, *CXCR3* expression reached its peak at the effector-memory state (Supplementary Fig. 6f). In peripheral T cells, *CXCR3* was expressed mostly in effector-memory T cells (Fig. 5h). Based on these findings, we hypothesized that the interaction between *CXCR3* and its ligands *CXCL9/10/11* plays an important role in recruiting peripheral CD8 effector-memory T cells (CD8 T$_{EM}$) into the TME. To confirm this, we again used CellChat to explore receptor-ligand interactions between the intra-tumoural macrophages and peripheral CD8 T cells. Indeed, taking all cell-cell communication networks into account, we found that CD8 peripheral T cells were the most dominant signalling 'receivers' in responders (Fig. 5i), with higher levels of incoming signals

compared to their phenotypical counterparts in non-responding tumours. Furthermore, within the CXCL-signalling pathway, there were more predicted interactions between intra-tumoural CXCL10$^+$ macrophages and peripheral CD8 T$_{EM}$ in responders compared to non-responders (Supplementary Fig. 6g).

Overall, this supports the notion that the intra-tumoural myeloid compartment, which is characterized by upregulated expression of *CXCL9/10/11* in responders, may be involved in the recruitment and activation of CXCR3$^+$ effector-memory T cells in the TME, potentially playing a role in determining response to atezo/bev.

### CD8 T$_{EMRA}$ and CXCL10$^+$ macrophages as predictive biomarkers of response to atezolizumab/bevacizumab in aHCC

Using scRNAseq, we identified clonally expanded, cytotoxic CD8 T$_{EMRA}$ as possible effector cells that drive response to atezo/bev, while CXCL10$^+$ macrophages (Macro CXCL10) might function as gatekeepers responsible for the recruitment of primed effector-memory peripheral T cells (Fig. 6). Next, we aimed to validate these single-cell derived findings and explore the potential of CD8 T$_{EMRA}$ and Macro CXCL10 as predictive biomarkers of response to atezo/bev in aHCC. Calculating a per sample CD8 T$_{EMRA}$ and Macro CXCL10 enrichment score in transcriptomic data of 311 pre-treatment tumour biopsies of aHCC patients treated with atezo/bev (n = 253) versus sorafenib (n = 58), we found that high CD8 T$_{EMRA}$ and Macro CXCL10 enrichment scores were associated with significantly longer PFS in atezo/bev-treated patients (Fig. 7a *top*), but not in sorafenib-treated patients (Fig. 7a *bottom*). Furthermore, the presence of CD8 T$_{EMRA}$ and Macro CXCL10 in the TME was strongly correlated (R = 0.84; p < 0.00001; Fig. 7b), supporting the notion that they populate the TME of aHCC together. Indeed, combining CD8 T$_{EMRA}$ and Macro CXCL10 marker genes into a single gene set, we found that atezo/bev-treated patients with a high enrichment score for the 'atezo/bev-response biomarker' had significantly longer OS and PFS (p = 0.049 and p < 0.0001, respectively), an association that was not seen in sorafenib treated patients (Fig. 7c). Taken together, the combined presence of CD8 T$_{EMRA}$ and Macro CXCL10 in the pre-treatment TME of aHCC patients is associated with improved outcomes upon atezo/bev treatment, specifically, validating the single-cell derived findings and underlining the potential value of the 'atezo/bev-response biomarker' as a predictive biomarker of response to atezo/bev in aHCC.

## Discussion

Our study represents the first, homogenous single-cell atlas of both the TME and peripheral immune system of aHCC patients treated with atezo/bev, allowing the correlation of single-cell readouts with durable and clinically-meaningful response.

Within the pre-treatment TME of aHCC, we found PD1-negative, CD45RA effector-memory CD8 T cells (CD8 T$_{EMRA}$) to be associated with response to atezo/bev. This clearly differs from other cancer types, where instead of CD8 T$_{EMRA}$, PD-1 expressing CD8 T$_{EX}$ have repeatedly been identified as key effector cells in response to CPI. Although both CD8 T$_{EX}$ and CD8 T$_{EMRA}$ contained clonally-expanded T cells, it was the CD8 T$_{EMRA}$, that were more abundant in the pre-treatment TME of responding tumours and found typical CD8 T$_{EMRA}$

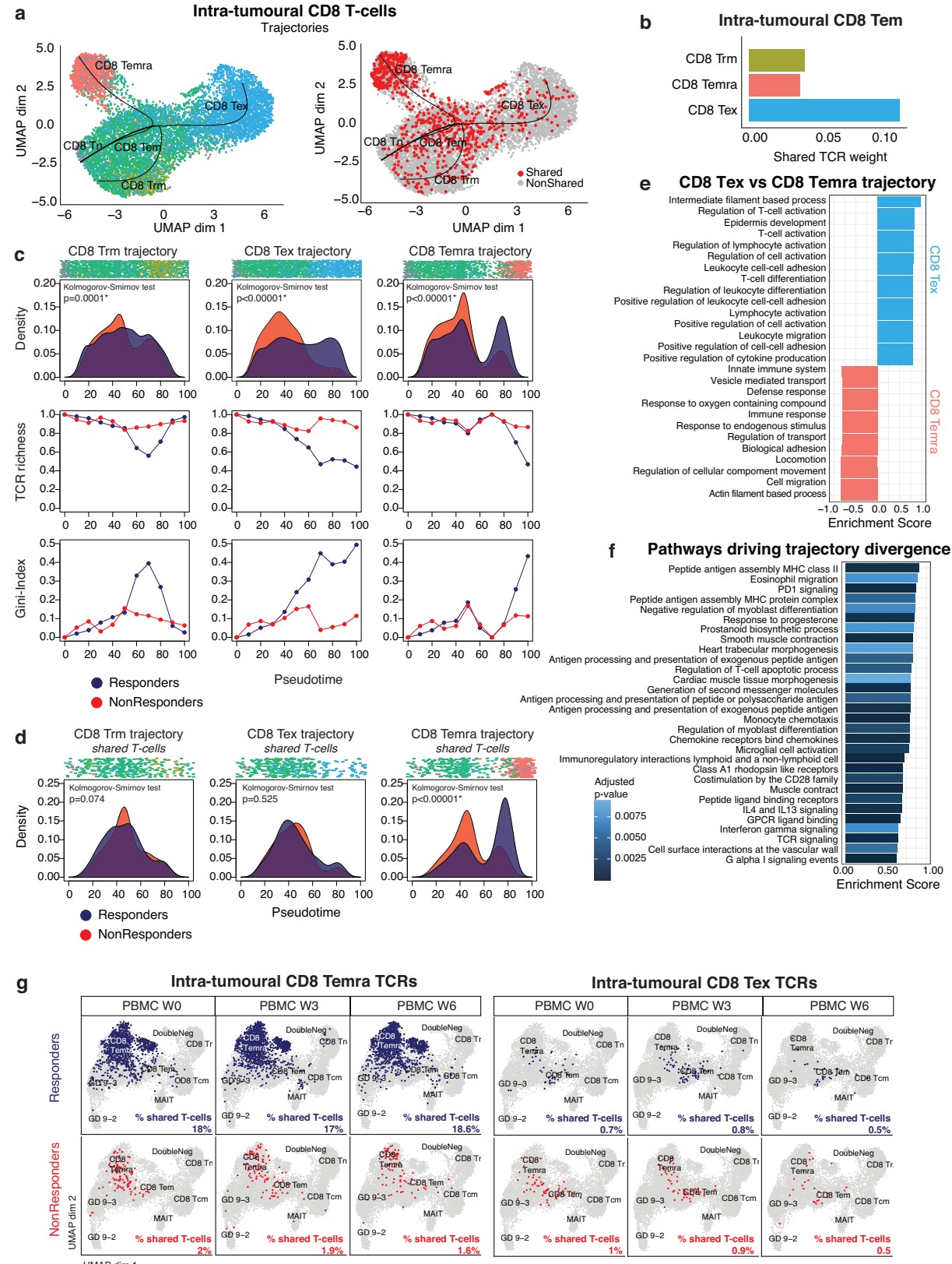

genes (*CX3CR1*, *SPON2* and *FCGR3A*) to be overexpressed in intra-tumoural T cells from responders compared to non-responders. Additionally, CD8 $T_{EMRA}$ displayed the highest degree of TCR sharing with peripheral blood, a phenomenon almost exclusively observed in responders which persisted on treatment, potentially suggesting that CD8 $T_{EMRA}$ are targeting tumour-specific antigens. Finally, using

trajectory analyses, we observed how intra-tumoural CD8 T cells in responders were capable of evolving towards more differentiated phenotypes, an effect most pronounced in the $T_{EMRA}$ trajectory, while non-responders seemed frozen at an earlier stage of the pseudo-time. T cells carrying a TCR shared between tumour and blood were also enriched towards the end of the $T_{EMRA}$ trajectory in responders,

**Fig. 4 | Interaction with tumour-antigens drives intra-tumoural differentiation towards CD8 T_EMRA.** **a.** UMAP representation of intra-tumoural CD8 T cells (n = 8989). *Left*: depicting three distinct trajectories: CD8 T_RM, CD8 T_EX and CD8 T_EMRA. *Right*: depicting shared T cells (i.e. intra-tumoural T cells characterized by a TCR found in PBMC week 0) versus non-shared T cells. **b.** Bar plot showing the shared TCR weight for CD8 T_EM with other CD8 phenotypes in the TME. **c.** T cell density, TCR richness and Gini-index along each CD8 trajectory in atezo/bev-treated patients (n = 20), stratified for response (12 Resp versus 8 NonResp). The density plots reflect the relative number of T cells separately for Resp versus NonResp along each CD8 trajectory. P-values reflect the difference in distributions, calculated using the two-sided Kolmogorov-Smirnov test. **d.** Density of shared T cells along each CD8 trajectory in atezo/bev-treated patients (n = 17) stratified for response (10 Resp versus 7 NonResp). The density plots reflect the relative number of shared T cells separately for Resp versus NonResp along each CD8 trajectory. Shared intra-tumoural T cells are characterized by a TCR found in peripheral blood prior to treatment. P-values reflect the difference in distributions, calculated using

the two-sided Kolmogorov-Smirnov test. **e.** Top 15 pathways identified using fGSEA on differentially expressed genes along CD8 T_EX versus CD8 T_EMRA trajectory (using *diffEnd* test; TradeSeq[35]) for the REACTOME and GO: Biological processes gene sets. Significantly enriched pathways (adjusted p-value < 0.01) were identified and ranked based on enrichment score for each gene set separately. Only the top 15 pathways of each gene set, enriched in each trajectory were retained. **f.** Top pathways identified using fGSEA on differentially expressed genes identified at the point where the three trajectories diverge (using *earlyDEG* test; TradeSeq[35]) for the REACTOME and GO: Biological processes gene sets. Adjusted p-value calculated using the Benjamini-Hochberg method. **g.** UMAP representation of peripheral T cells at week 0 (W0), week 3 (W3) and week 6 (W6) characterized by a TCR shared with CD8 T_EMRA (*left*) and CD8 T_EX (*right*) in the pre-treatment TME, stratified for response to atezo/bev (*top*: Resp (n = 10), *bottom*: NonResp (n = 17)). (PBMC, peripheral blood mononuclear cells; TCR, T cell receptor; TME, tumour microenvironment; UMAP, Uniform Manifold Approximation and Projection; Resp, responder; NonResp, non-responder).

specifically. In contrast, PD1-expressing CD8 T_EX cells were not more abundant or more clonal in responders compared to non-responders. Along the T_EX trajectory, the greatest T cell density was seen at the T_EM stage both in responders and non-responders and their TCRs were not found in blood prior to treatment, nor did they appear during therapy. This suggests that against the backdrop of the immunosuppressive milieu of the liver, PD1-expressing CD8 T cells do not become activated during response to CPI, as they do in other cancer types[18,20].

Certainly, it seems that CD8 T_EMRA are able to overcome immunosuppression within the liver TME as our findings point towards CD8 T_EMRA as the main candidate effector cell type of anti-tumoural immunity upon atezo/bev treatment in aHCC. While CD8 T_EMRA have been previously identified in the TME of early stage HCC patients[13,14,17], their role in response to systemic therapy has never been described. Interestingly, CD8 T_EMRA are more abundant and more clonally-expanded in aHCC compared to our observations in other cancer types[18,44]. Notably, CD8 T_EMRA do not express the typical exhaustion markers associated with activation and antigen-experience, nor do they express markers associated with a TME enriched for high cytokine expression or marked interferon gamma signalling. Instead, they re-express CD45RA upon antigen stimulation and are characterized by an NK-like functional phenotype, endowed with potent cytolytic properties that are mediated by the release of lytic granules and rely on direct interaction with target cells.

Nonetheless, the PD1-negative status of CD8 T_EMRA suggests that they may not be the direct therapeutic targets of atezo/bev. Indeed, within the myeloid compartment, we identify activated, pro-inflammatory, PDL1-expressing CXCL10+ macrophages as potential regulators. Analogously to the suppressive role of PD1 in T cells, PDL1 is an inhibitory activation marker for macrophages, designed to prevent uncontrolled inflammation[45]. Pre-clinical research has shown that upon treatment with CPI, PDL1-expressing myeloid cells proliferate and are activated[45,46]. In line with this, we found that CXCL10+ macrophages in atezo/bev-responders express higher levels of PDL1 and this was associated with better outcomes upon treatment, suggesting that atezo/bev treatment may lead to increased activation of PDL1-expressing CXCL10+ macrophages, releasing their chemokines (*CXCL9/10/11*) into the TME. The importance of *CXCL9/10/11* in the therapeutic efficacy of CPI, by their role in T cell recruitment, has been previously described in other cancer types[47–49]. Intriguingly, *CXCR3*, the main target of *CXCL9/10/11*, was expressed predominantly in peripheral T_EM and along the T_EMRA trajectory *CXCR3* expression reached its peak during the effector-memory phase. This suggests that increasing CXCL10+ macrophage activity may lead to more efficient and continued peripheral T cell recruitment, replenishing the intra-tumoural CXCR3+ T_EM population. Subsequently, within the tumour and upon antigen stimulation, these CXCR3+ T_EM

preferentially differentiate towards PD-1 negative CD8 T_EMRA, the proposed effectors of direct anti-tumour cytotoxicity within the TME. Indeed, the presence of CD8 T_EMRA and Macro CXCL10 in the pre-treatment TME was strongly linked and our data suggest that their combined presence is associated with improved outcomes upon atezo/bev-treatment, specifically, advocating for their potential value as predictive biomarkers of response to atezo/bev in aHCC.

Our findings also highlight key questions for future research. Firstly, in line with their patrolling phenotype, CD8 T_EMRA display significant TCR sharing with peripheral blood, particularly in atezo/bev-responders and TCR sharing was associated with improved PFS. Whether this peripheral CD8 T_EMRA population is identifiable prior to therapy remains to be confirmed, providing a unique opportunity for exploration of their potential as non-invasive biomarkers. A second question relates to the molecular signals that mediate the preferential differentiation of CD8 T_EM towards CD8 T_EMRA. HCC most often develops in a background of chronic inflammation that eventually leads to cirrhosis and malignant transformation. This unique inflammatory milieu may be the ideal ground to attract CD8 T_EM into the liver via *CXCR3*. We identified antigen presentation pathways as elements that may influence whether CD8 T_EM cells then subsequently differentiate towards clonally-expanded T_EMRA or not, but these observations require further validation and the exact underlying molecular signals still need to be identified. Finally, characterizing the signals that influence the differentiation from CD8 T_EM to CD8 T_EX could offer novel therapeutic targets to stimulate the differentiation of anti-tumoural CD8 T_EX cells towards a more activated phenotype, particularly in those patients displaying primary resistance to atezo/bev.

There were a number of limitations associated with our study, including the lack of on-treatment tissue biopsies, which are notably difficult to obtain in this setting. Collecting tissue biopsies from aHCC patients *prior* to treatment is already quite exceptional, as they are not always feasible in a fragile aHCC patient population and, though recommended, not mandated by clinical guidelines. As a result, we cannot capture the effect of atezo/bev within the tumour and are potentially missing emerging new dimensions of anti-tumour immunity. Additionally, due to a limited number of viral aHCC samples, our study does not allow us to draw conclusions related to the role of viral versus non-viral aetiologies and response to atezo/bev, a much debated concept that warrants further research. We validated the association of the intra-tumoural presence of CD8 T_EMRA and CXCL10+ macrophages with overall and progression free survival in a publicly available bulk RNAseq dataset. Additional patient or tumour characteristics were not available to perform a multivariate cox regression analysis. Finally, due to their descriptive nature, single-cell studies are appropriate to

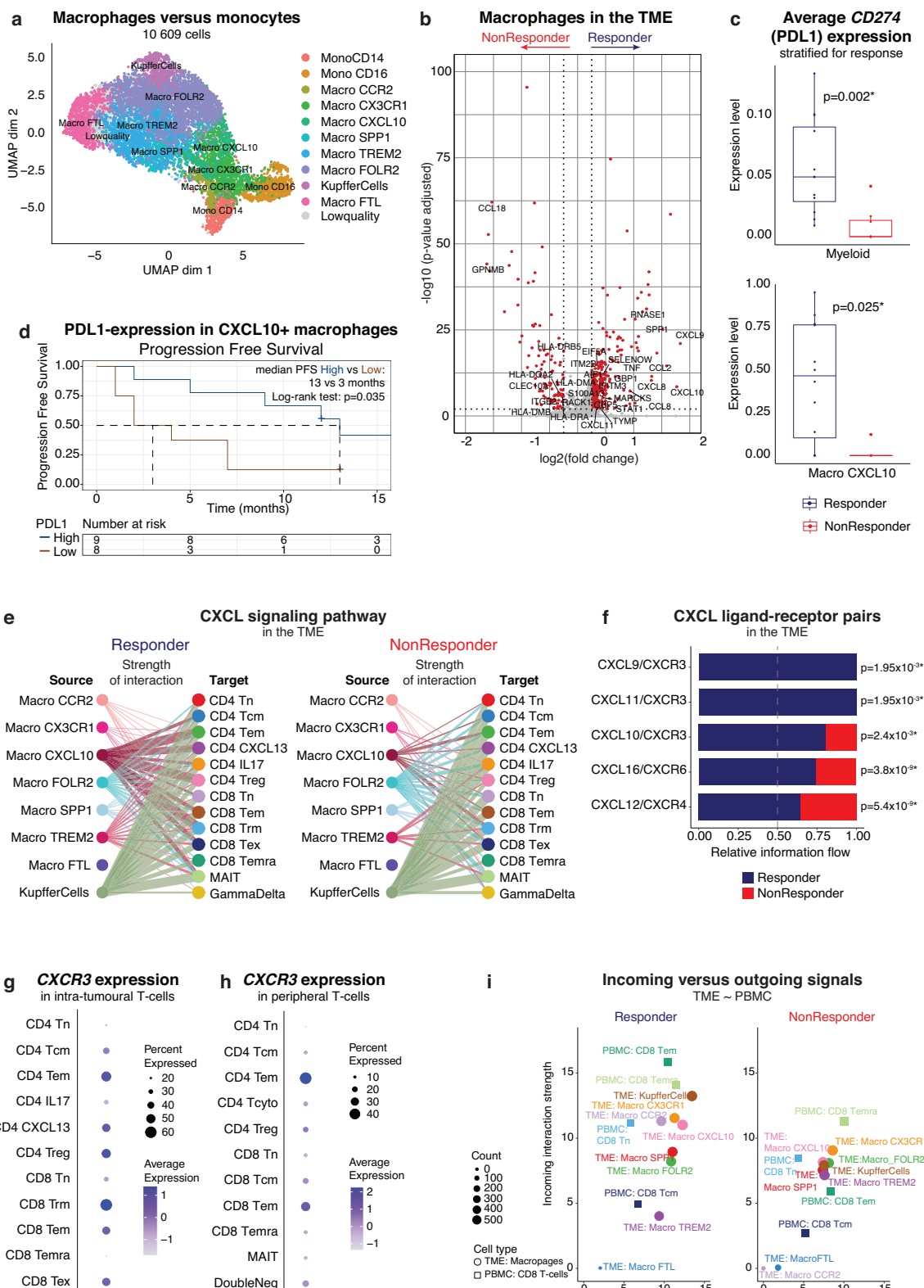

explore associations. Demonstrating causation would require functional validation studies.

In conclusion, we propose a novel paradigm, where response to atezo/bev in aHCC is driven by clonally expanded, cytotoxic CD8 $T_{EMRA}$ characterized by a high degree of TCR sharing with peripheral blood and present in the tumour prior to therapy. PDL1-expressing CXCL10+

macrophages are positioned as essential gatekeepers in the TME, interacting with the peripheral T cell compartment to ensure effective T cell recruitment into the TME. While the single-cell resolution was essential for explorative purposes, we demonstrate the predictive value of CD45RA effector-memory CD8 T cells and CXCL10+ macrophages as biomarkers of response to atezo/bev in aHCC using bulk RNAseq.

**Fig. 5 | Pro-inflammatory PDL1-expressing CXCL10+ macrophages recruit effector-memory T cells into the TME. a**. UMAP representation of monocyte and macrophage phenotypes in the TME. **b**. Volcano plot of differentially expressed genes in macrophages (n = 9233) in the TME of responders (n = 12) versus non-responders (n = 8). P-values were obtained using the two-sided Wilcoxon test and Bonferroni-corrected (Seurat 4[53]). *Red*: adjusted p-value < 0.01 and log2 fold change >0.25. **c**. Boxplots depicting average *CD274* (PDL1) expression level in the TME of atezo/bev-treated patients, calculated per patient in myeloid cells (*top;* n = 20; 12 Resp versus 8 NonResp) and CXCL10+ macrophages (Macro CXCL10, *bottom;* n = 17; 12 Resp versus 5 NonResp), stratified for response. P-values calculated using two-sided Mann-Whitney U-test. Boxes indicate median +/- interquartile range; whiskers show minima and maxima. **d**. Kaplan-Meier plot of progression free survival in atezo/bev-treated patients (n = 17) with high or low (split by median) PDL1-expression in CXCL10+ macrophages. **e**. Hierarchy plot of the CXCL signalling pathway in the TME, depicting cell-cell interactions between intra-tumoural

macrophages (source) and intra-tumoural T cells (target cells) in responders (*left*) and non-responders (*right*). The width of edges represents the strength of communication. **f**. Bar plot depicting the overall information flow for each ligand-receptor pair of the CXCL signalling pathway in the TME of responders versus non-responders. Overall information flow is defined by the sum of the communication probability among all pairs of cell groups in the inferred network. P-value calculated using two-sided Wilcoxon test. **g**. Dot plot of *CXCR3* expression in intra-tumoural T cells. **h**. Dot plot of *CXCR3* expression in peripheral T cells. **i**. Scatterplot depicting the dominant senders (sources) and receivers (targets) between intra-tumoural macrophage phenotypes and peripheral CD8 T cells. X- and y-axis represent the total outgoing or incoming communication probability associated with each cell group. Symbol size is proportional to the number of inferred links (both outgoing and incoming) associated with each cell group. (PBMC, peripheral blood mononuclear cells; PDL1, Programmed death-ligand 1; TME, tumour-microenvironment; UMAP, Uniform Manifold Approximation and Projection).

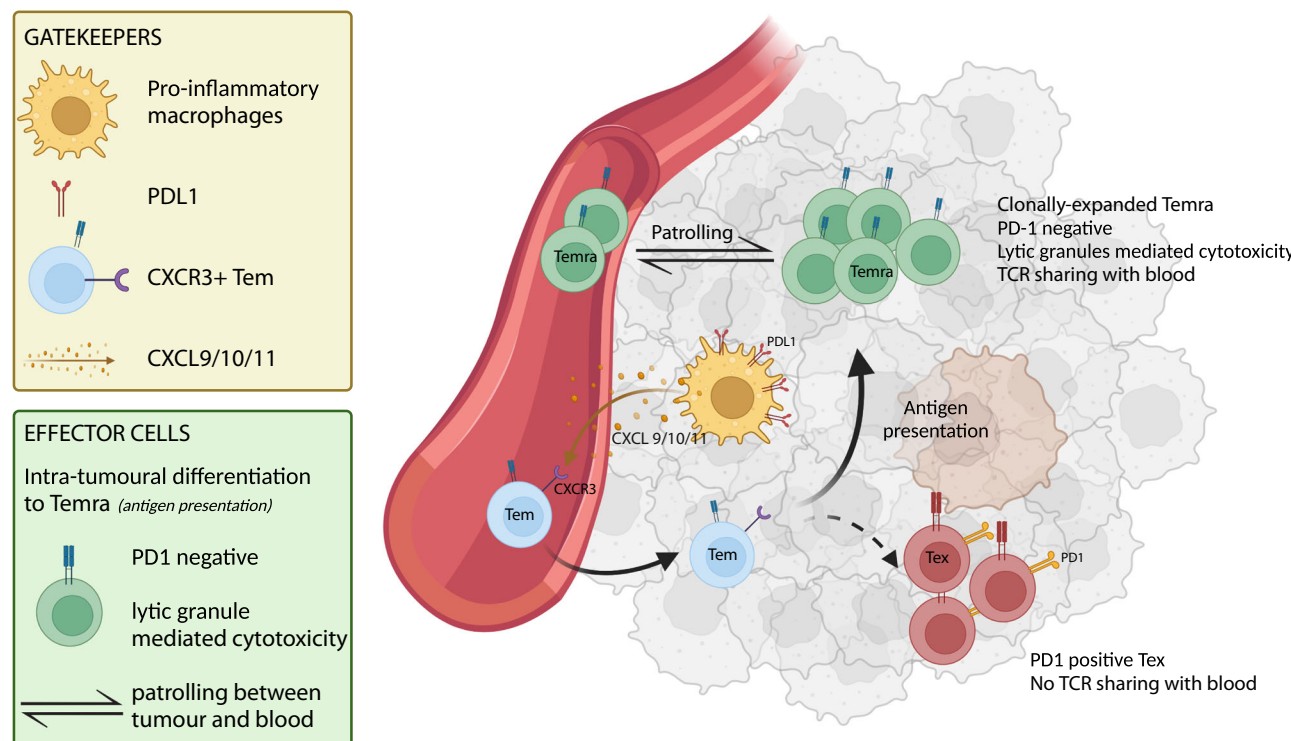

**Fig. 6 | CD8 T_EMRA and Macro CXCL10 play an essential role in response to atezolizumab/bevacizumab in advanced HCC.** Graphical representation of CD8 T_EMRA and Macro CXCL10 in the TME of advanced HCC and their potential role in facilitating response to atezo/bev. (PD1, Programmed cell death protein 1; PDL1, Programmed death-ligand 1; TCR, T cell receptor; TME, tumour-microenvironment). Created using Biorender.com.

## Methods

### Inclusion and Ethics

The study was approved by the Ethics Committee of University Hospitals Leuven (UZ/KUL, S62548). All patients gave written informed consent for the use of their samples for research purposes.

### Patients and methods

Between December 2018 and June 2023, all patients diagnosed with aHCC and eligible for systemic treatment at the University Hospitals Leuven, were invited to participate in our study. Clinical eligibility was based on good performance status (ECOG 0-1) and adequate haematologic and end-organ function. Selection of systemic treatment was at the discretion of the treating physician, guided by clinical practice guidelines, individual patient eligibility and treatment availability at time of inclusion. Radiological response was evaluated by computed tomography (CT) or magnetic resonance

imaging (MRI) approximately every 3 months, according to standard clinical practice and assessed by an independent radiologist using the modified RECIST criteria[50]. Response was defined as objective response (partial or complete response) at 3 months or disease control (stable disease) during at least 6 months after treatment initiation.

Prospective sample collection included a fresh tissue biopsy before start of treatment and serial PBMC samples collected prior to and during treatment (week 0-3-6). Overall 38 tissue biopsies and 72 PBMC samples were available, an overview is provided in Supplementary Table 2. For two patients, two biopsies from the same tumour nodule were taken. All samples were subjected to simultaneous scRNAseq and scTCRseq, as previously described[18,44,51]. scRNAseq data from all available samples was used for clustering and annotation of single cells into their respective tumoural and peripheral cell phenotypes. As we specifically aimed to explore the

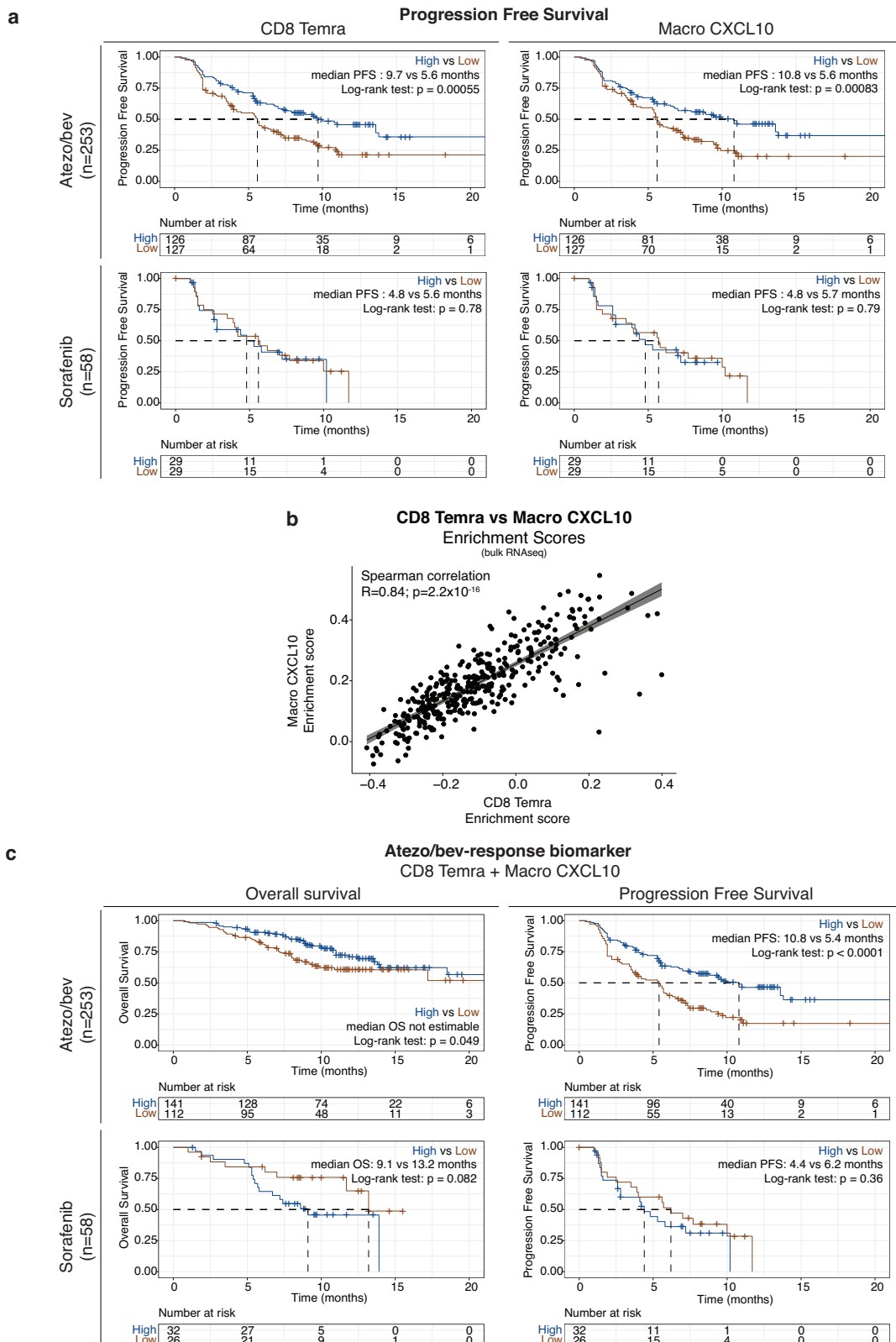

effect of atezo/bev, all subsequent comparative analyses focussed on atezo/bev-treated patients (n = 25), stratified according to clinical response (15 responders versus 10 non-responders). For all comparative analyses, response to atezo/bev was the primary endpoint, while progression free survival was considered a secondary endpoint.

**Sample collection and processing**
Fresh tissue biopsies (n = 38) were obtained via diagnostic needle biopsy with 18-G needles prior to start of systemic therapy and immediately subjected to single-cell dissociation as previously described[18,44,51]. The tissue samples were first mechanically dissociated using a scalpel, followed by enzymatic dissociation in digestion

**Fig. 7 | CD8 T_EMRA and Macro CXCL10 as predictive biomarkers of response to atezolizumab/bevacizumab. a**. Kaplan-Meier plots of progression free survival in 253 atezo/bev-treated (*top*) versus 58 sorafenib-treated advanced HCC patients (*bottom*) with high versus low (split by median) CD8 T_EMRA (*left*) or Macro CXCL10 (*right*) enrichment score, calculated per sample in bulk RNA-sequencing data. P-values calculated using the log-rank test. Median progression free survival as indicated. **b**. Scatter plot depicting the correlation between CD8 T_EMRA and Macro CXCL10 enrichment scores, calculated per sample in bulk RNA-sequencing data of 311 advanced HCC patients. P values calculated using two-sided Spearman's rank correlation test. R-values are Spearman's rank correlation coefficients (rho). Error bands represent the 95% confidence interval. **c**. Kaplan-Meier plots of overall survival (left) and progression free survival (*right*) in 253 atezo/bev-treated (*top*) versus 58 sorafenib-treated advanced HCC patients (*bottom*) with high versus low atezo/bev-response biomarker enrichment score (using optimal cut-off determined by maximally selected rank statistics, see Methods for details), calculated per sample in bulk RNA-sequencing data. (Atezo/bev, atezolizumab plus bevacizumab; PFS, Progression free survival).

medium (2 mg ml⁻¹ Collagenase P (Sigma Aldrich) and 0.2 mg ml⁻¹ DNAse I (Roche) in DMEM (Thermo Fisher Scientific)). Red blood cells were removed from the cell suspension using red blood cell lysis buffer (Roche), and the cells were filtered using a 40-μm Flowmi tip strainer (VWR). The number of living cells was determined using a LUNA automated cell counter (Logos Biosystems). Peripheral blood mononuclear cells (PBMCs) were extracted from serial peripheral blood samples (n = 72) by immunomagnetic negative selection using 'EasySep™ Direct Human PBMC Isolation Kit' (Stemcell Technologies). Red blood cells were removed using red blood cell lysis buffer (Roche). Cells were filtered using a 40-μm Flowmi tip strainer (VWR), the number of living cells was determined using a LUNA automated cell counter (Logos Biosystems) and stored at −80 °C (in FBS + 10% DMSO) for simultaneous thawing at a later timepoint.

PBMC samples were thawed simultaneously by adding DMEM stepwise, cells were filtered using 40-μm Flowmi tip strainer (VWR) and the number of living cells was counted using a LUNA automated cell counter (Logos Biosystems). Up to 1 million cells from two or three different samples were pooled together in equal proportions. The pooling matrix was designed in such a way to allow for bio-informatic identification of samples after sequencing (see below for details). Simultaneous epitope measurement was performed on 41 PBMC samples. First, the cells were incubated on ice with 5 μl of Fc receptor blocking solution (TruStain Fcx from Biolegend) for 10 minutes. Next, 15.9 μl of TotalSeq-C (Biolegend) 162 antibody-oligo pool (1:1,000 diluted in labelling buffer (PBS + 1%BSA) until 100 μm; full list in Suppl. Dataset 1), followed by another 30-min incubation on ice. Cells were washed three times with labelling buffer and filtered through a 40-μm Flowmi strainer.

### Single-cell RNA-sequencing, T cell repertoire profiling and cell surface epitope profiling

We performed single-cell TCR-seq and 5' gene expression profiling on the single-cell suspensions derived from fresh tissue biopsies and PBMC samples using Chromium Single Cell V(D)J Solution from 10X Genomics according to the manufacturer's instructions. Up to 5000 cells per sample for biopsies and up to 20 000 live cells for pooled PBMC samples were loaded onto a 10X Genomics chip to generate cell-barcoded 5' gene expression libraries. For two patients, we obtained two biopsies from the same tumour nodule for which separate libraries were generated. The libraries were sequenced on an Illumina NextSeq and/or NovaSeq6000 system, and mapped to the GRCh38 human reference genome using Cell-Ranger (10x Genomics). V(D)J enriched libraries were sequenced on an Illumina HiSeq4000 system and TCR alignment and annotation were achieved with CellRanger VDJ (10x Genomics; Version 3.1.0). Additional epitope profiling was performed by TotalSeq-C (Biolegend) on a subset of PBMC samples (n = 41). These samples were processed as described above, with the addition of a separate library of barcode-tagged antibodies for each cell. The RNA-derived 'Gene Expression library' was mapped to the GRCh38 human reference genome using CellRanger (10x Genomics) as described above, while the protein-derived 'Antibody Capture library' was mapped to the whole TotalSeq-C antibody list.

### PBMC patient-ID assignment using Souporcell

As described above, PBMC samples were pooled, loading two or three samples per lane in the 10X Genomics chip in equal proportions. The Souporcell tool[52] was used to assign each cell back to its sample of origin. In short, the tool first remaps the scRNAseq data of the input samples using the Minimap2 mapper. The remapped data is then analysed for variants on a per cell basis, followed by clustering based on co-occurring variants to assign each cell a probability of belonging to each of the clusters. Cells carrying co-occurring variants are assigned to a sample specific cluster. The pooling matrix was designed in such a way that each cell cluster in Souporcell could easily be linked back to the corresponding sampleID. The design of the pooling matrix is represented in Supplementary Table 3.

### Single-cell gene expression analysis (scRNAseq) of tumour biopsies

Raw gene expression matrices generated by CellRanger (10x Genomics) were analysed further using Seurat 4[53] using default parameters unless otherwise specified. All samples were merged into a single Seurat object. Barcodes expressing <200 and >6000 genes and <400 unique molecular identifiers (UMIs) were removed. All cells with >50% mitochondrial RNA content were removed as they likely represent dying cells. Previous single-cell studies in liver and liver cancer have used varying cut-off from 30%[54] up to 50%[17,55]. Gradually decreasing the threshold from 50% to 30% primarily removed annotated cancer cells or low-quality cells in downstream analysis, leaving the immune cells unchanged.

A total of 97 947 cells ([58-7311] cells per sample) with on average 1320 genes per cell and 4676 unique transcripts per cell were retained and normalized (using *NormalizeData* function). The 2000 most variable genes were identified (using *FindVariablesFeatures* function) and principal component analysis (PCA) was applied to reduce dimensionality after regressing for the number of UMIs, percentage of mitochondrial RNA and cell cycle genes (S and G2M scores, calculated using *CellCycleScoring* function). The 25 most informative principal components (PCs) were retained for clustering and Uniform Manifold Approximation and Projection for dimension reduction (UMAP). However, the resulting UMAP revealed clustering based on sample-specific variation. To correct for this, the Harmony[56] algorithm was applied to regress out sample-specific effects (using the first 25 PCs), resulting in a well-integrated dataset (Supplementary Fig. 7). As expected, malignant cell clusters were patient-specific, while non-malignant clusters contained cells derived from different patients (Supplementary Fig. 7). There was no evidence of cluster bias based on underlying liver disease, treatment or biopsy type in immune cells and stromal cell types (Supplementary Fig. 1d). The data were clustered using *FindNeighbours* and *FindClusters* functions. The resulting two-dimensional UMAP representation of the dataset consisted of distinct major cell types, identified and annotated based on the expression of known marker genes.

### scRNAseq clustering leading to cell subtypes in tissue biopsies

To subcluster T- and NK-cells into their respective phenotypes, we subset T-/NK-cells annotated at the major cell type level. We applied

the same process as described above, with an additional removal of TCR genes prior to the identification of variable features, in order to avoid clustering based on TCR genes. We first used marker genes to identify CD4+ T cells, CD8+ T cells, NK-cells and proliferating cells. Subsequently, applying an identical process, we subclustered the CD4+ T cells, CD8+ T cells, and proliferative cluster separately into their cellular (sub-)phenotypes. Finally, all subsets were merged back into one annotated T-/NK-cell Seurat object for further downstream analyses. Similarly, myeloid cells were annotated into dendritic cells versus monocytes/macrophages. Dendritic cells were merged with plasmacytoid dendritic cells for subclustering and annotation. Monocytes and macrophages were subclustered separately into their respective phenotypes.

## Single-cell copy number analysis in tumour biopsies

Copy number variations (CNV) were assessed with the R package inferCNV[25], designed to infer CNVs from tumoural scRNAseq data. InferCNV compares the expression of genes in malignant cells to the expression in cells annotated as non-malignant. T-/NK-cells, B-cells and myeloid cells were used as a reference for non-malignant cells.

## Integration, clustering and annotation of scRNAseq and TotalSeq-C data in PBMC samples

Raw gene expression matrices from the PBMC samples were generated using CellRanger 3.1 (10x Genomics) and analysed using Seurat 4[53]. One Seurat object was generated with the scRNA-seq data and the antibodies present in the antibody pool (Supplementary Dataset 1). All barcodes expressing <200 and >6000 genes, <400 UMIs and >15% mitochondrial DNA content were removed. Next, we used the Souporcell clusters to link each barcode to its sample of origin. All cells classified with insufficient confidence were removed. A total of 268 807 cells ([240-8528] cells per samples) with on average 1223 genes per cell and 3235 unique transcripts per cell were retained.

The PBMC subset with only scRNAseq data available was processed similarly to the scRNAseq data from pre-treatment biopsies. For PBMC samples with RNA and antibody data available all features were reported as variable features. The data were normalized by Centred-Log-Normalisation (CLR) using the second margin (using *NormalizeData)* and subsequently scaled for ADT-count. Finally, dimensionality reduction was performed using PCA and the UMAP representation was generated.

Next, the RNA- and ADT-assays were combined using a 'weighted nearest neighbour' analysis[53]. In short, a new 'integrated' assay was generated using *FindMultiModalNeighbours* function by assigning a weight to each cell based on the relative contribution of the RNA- versus ADT-assay to the clustering process. A new UMAP was generated using the integrated dataset and clustering analysis was performed (using *FindCluster* function with the Smart Local Moving algorithm).

Subsequently, the PBMC subset without ADT-data was projected onto the 'integrated' assay. Combining both the RNA- and ADT-assay, allowed for a better biological separation of cells based on ADT-data (e.g., CD4+ versus CD8+ T cells). Manual annotation was performed iteratively based on marker gene expression. First, the major peripheral cell types were annotated, followed by annotation of peripheral T-/NK-cells into their respective phenotypes.

## Differential gene expression and pathway analysis

Differentially expressed genes (DEG) were identified using Wilcoxon's test with the *FindMarkers* function from Seurat without a threshold for log fold-change (logFC) and for expression in a minimum fraction of cells. DEGs were then ordered based on average log fold-change and used as input for GSEA using the R package f-GSEA for the Hallmarks of the cancer gene set. The resulting list of enriched gene sets was filtered for adjusted *p*-values < 0.01 (using Benjamini–Hochberg method).

## Assessing the TCR repertoire using V(D)J analysis

In the biopsies, we detected 19,220 out of 26 380 T cells with a productive TCR sequence, meaning that they could be joined in the proper reading frame by V(D)J recombination without a premature stop codon, enabling the expression of a complete TCR alpha or beta-chain for downstream analysis. Excluding NK-cells, gamma-delta T cells, and MAIT cells (restrictive TCR), 90% of annotated T cells annotated carried a productive TCRs, resulting in a total of 19 818 annotated T cells with a productive TCR, carrying a total of 12 690 unique TCRs. Of note, one sample (HCC006) was removed from further TCR analysis, as we did not detect annotated T cells carrying productive TCRs.

In PBMCs, we detected 121 765 out of 170 919 T-/NK-cells carrying a TCR sequence. Again, we considered only productive TCR sequences. Excluding NK-cells, gamma-delta T -cells and MAIT cells (restrictive TCR), 90% of peripheral T cells carried a productive TCRs. A total of 25 unique TCR sequences were shared between at least two patients and were removed for further analysis. This resulted in a total of 115,711 annotated peripheral T cells, carrying 90,188 unique TCRs.

Next, TCR clonotypes were defined as TCRs with the same complementarity-determining region 3 (CDR3) nucleotide sequences. Dominant clonotypes were defined as 1) TCRs shared by 5 or more T cells and 2) clonotypes representing at least 1% of the TCR repertoire in each sample. Clonality was defined as the complement of evenness (1-evenness), as previously described[18,57], where evenness represents the normalized Shannon entropy. The evenness value lies between 0 and 1, with a high value indicating a more equal distribution of TCRs and a low value indicating TCR skewing due to clonal expansion. TCR richness was defined as the number of unique TCRs divided by the total number of cells with a unique TCR and was calculated as a metric for clonotype diversity. Finally, the Gini-index was calculated using the *ineq* (v0.2-13) package in R and captures the distribution of T cells across the TCR repertoire. The value ranges between 0 and 1. The higher the Gini-index, the less equal the distribution of the clonotypes. Each TCR metric (clonality, evenness, richness, Gini-index) was calculated per sample and then per T cell phenotype in each sample.

## TCR sharing between intra-tumoural T cells and peripheral T cells

For 19 patients both pre-treatment tumour biopsies and serial PBMC samples were available (Supplementary Table 2), of which 17 were treated with atezo/bev. In this overlap-dataset, we detected a total of 112 810 annotated T cells, carrying 80 363 unique TCRs. A total of 442, 464, and 339 TCRs were shared between biopsy and PBMC at week 0, week 3, and week 6, respectively. The proportion of shared TCR in peripheral blood was calculated by dividing the number of shared TCRs between pre-treatment biopsy and PBMCs by the total number of TCRs detected in each sample. Similarly, the proportion of shared peripheral T cells was calculated by dividing the number of Tcells carrying TCRs shared between pre-treatment biopsy and PBMCs by the total number of Tcells carrying a productive TCR in each sample.

## Trajectory inference analysis

The R package Slingshot v2.2.0[34] was used to define computationally imputed pseudotime trajectories of CD8 T cells in the TME. Naive T cells (CD8 $T_N$) were considered as the root state when calculating the trajectories and the pseudotime. To visualize the degree of overlap in TCR clonotypes between T cell phenotypes, the connection weight for each pair of T cell phenotypes was calculated as the shared number of unique TCRs divided by the total number of unique TCRs in the T cell phenotype located first on the trajectory (=shared TCR weight). We then used TradeSeq v1.7.07[35] to identify DEGs between trajectories. We first used the *diffEnd* function, a between-lineage test to identify DEGs between the terminally differentiated ends of each trajectory. Then, we used the *earlyDEG* function using knots 1-6, to assess the differences in

expression patterns early on in the CD8 trajectories. The DEG lists were then used as input for GSEA using the R package f-GSEA for the REACTOME and GO: Biological Processes gene sets. Only significantly enriched pathways were retained (adjusted $p$-value < 0.01).

## Predicting cell-to-cell interactions in scRNAseq data

The CellChat (v1.1.3) algorithm[43] was used to predict cell-cell interactions between cell types in scRNAseq data, using default parameters with the following exceptions: the number of permutations used was 10 000 and cell-cell interactions between cell types were not considered when less than 15 cells represented a group. We focused on significant cell-cell interactions between immune cell types in responders and non-responders, separately ($p$-value < 0.01).

## Estimating progression free survival in the single-cell cohort

Progression free survival (PFS) was defined as the time from the start of atezo/bev to documented radiological progression (according to mRECIST) or death of any cause. The progression free survival probability was calculated for two features associated with response to atezo/bev: (1) TCR sharing between tumour and peripheral blood (PBMC Week 0) and (2) PDL1-expression in CXCL10+ tumour-associated macrophages (Macro CXCL10). For each metric, patients were grouped into two groups compared to the median (high versus low). The Kaplan-Meier method was used to estimate PFS curves and the log-rank test was used to assess significant differences between groups. Considering the limited number of events in the scRNAseq cohort, the PFS analysis was not corrected for baseline patient or tumour characteristics in a multivariate analysis. Analyses were done using the R packages 'survival' (version 3.3.1) and 'survminer' (version 0.4.9).

## Validating single-cell findings in bulk RNA seq dataset of 311 advanced HCC patients

In order to validate single-cell derived insights and explore the potential of CD8 $T_{EMRA}$ and Macro CXCL10 as predictive biomarkers of response to atezo/bev, we used CD8 $T_{EMRA}$ and Macro CXCL10 marker genes in order to deconvolute a publicly available bulk RNAseq dataset[24] (EGAS0001005503; DA00468). Overall, 311 prospectively collected tumour samples of advanced HCC patients treated with atezo/bev ($n = 253$) or sorafenib ($n = 58$) in the context of the phase Ib (GO30140; arms A and F)[11] and phase III clinical trials (IMBrave150)[3,4] were used. For details on study design and patient cohorts refer to the original publication[24]. The per sample raw RNA read files and associated clinical data were downloaded from the European Genome Archive (EGAS0001005503; DA00468). The raw read files were mapped to the human reference genome (refdata-gex-GRCh38-2020-A) using the STAR aligner (STAR.2.7.2a) in paired-end mode. Gene counts per sample were then computed using featureCounts (Subread toolkit) and the RNA counts were normalized based on the trimmed mean of M-values (TMM) method using the R-package edgeR (version 3.3.2). The resulting effective library size was used for downstream analysis.

CD8 $T_{EMRA}$ ($n = 36$) and Macro CXCL10 ($n = 16$) marker genes were selected based on differential gene expression analysis and used to deconvolute bulk RNAseq data. The CD8 $T_{EMRA}$ gene signature consisted of "GZMH", "GNLY", "NKG7", "FGFBP2", "GZMB", "CST7", "CCL5", "PRF1", "CX3CR1", "CTSW", "GZMA", "KLRD1", "GZMM", "CD3D", "CD8A", "CD52", "PTPRC", "CD3G", "HCST", "CD3E", "PLEK", "KLRG1", "RAC2", "LCK", "CD247", "HOPX", "KRLK1", "BIN2", "S100A4", "CORO1A", "IL2RG", "ITGB2", "IFITM1", "EMP3", "TRBC1", "FLNA". The Macro CXCL10 signature consisted of "CXCL10", "CXCL9", "GBP1", "TYMP", "CALHM6", "CCL2", "TNFSF13B", "WARS", "CCL8", "IL4I1", "ICAM1", "LILRB4", "CXCL11", "SOD2", "LAP3", "STAT1". Each gene set was used to calculate per sample enrichment scores for CD8 $T_{EMRA}$ and Macro CXCL10,

respectively using single-sample Gene Set Enrichment Analysis (ssGSEA) function from the R-package 'GSVA' (version 1.38.2). For each cell type, samples were divided into two groups: high versus low enrichment score (split by median) and the Kaplan-Meier method was used to estimate and compare PFS between groups using the log-rank test. Finally, we combined the CD8 $T_{EMRA}$ and Macro CXCL10 markers genes into an atezo/bev response biomarker, comprising a total of 52 genes. Calculating the per sample bulk RNAseq enrichment score (ssGSEA), two groups with high (biomarker^high) versus low (biomarker^low) enrichment score for the atezo/bev response biomarker were delineated using maximally selected rank statistics as described in the R package 'maxstat' (version 0.7.25). Kaplan-Meier curves for OS and PFS were generated for patients treated with atezo/bev ($n = 253$) versus sorafenib ($n = 58$). Significant differences between groups (biomarker^high vs biomarker^low) were evaluated using the log-rank test. Patient and tumour characteristics were not publicly available to perform a multivariate cox regression analysis for PFS. All analyses were done using the R packages 'survival' (version 3.3.1) and 'survminer' (version 0.4.9).

## Reporting summary

Further information on research design is available in the Nature Portfolio Reporting Summary linked to this article.

## Data availability

Raw sequencing reads of the scRNAseq, scTCRseq, and Totalseq experiments are available with restricted access in the European Genome-phenome Archive under accession number EGAS00001007547. Requests for accessing raw sequencing reads will be reviewed by the UZLeuven-VIB data access committee. Any data shared will be released via a Data Transfer Agreement that will include the necessary conditions to guarantee the protection of personal data (according to European GDPR law). Alternatively, a download of the read count data per sample, necessary to reproduce all analyses included in this manuscript, will be made available at https://lambrechtslab.sites.vib.be/en/data-access. Source data are provided in this paper.

The per sample raw RNA read files from the publicly available bulk RNAseq dataset and associated clinical data were downloaded from the European Genome Archive (EGAS00001005503; DA00468; https://ega-archive.org/studies/EGAS00001005503). Source data are provided in this paper.

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

## Acknowledgements

The computational resources and services used in this work were partly provided by the Flemish Supercomputer Centre (VSC), funded by the Research Foundation—Flanders (FWO) and the Flemish Government department EWI. We thank E. Vanderheyden for technical assistance. This study was supported by the 'Stichting Tegen Kanker' (FAF-C/2018/1297) grant. S.C. was supported by a strategic basic research fellowship from the Research Foundation—Flanders (FWO; 1S95221N). D.L. was supported by an ERC Advanced Grant (101055422) and KU Leuven internal fund (C14/18/092).

## Author contributions

D.L. and J.D. jointly supervised this work. D.L. designed and supervised the single-cell experiments. J.D. designed the clinical study and supervised sample collection and clinical annotation with important help from S.C., as well as C.V., E.V.C., J.J., H.T., B.T., and V.V. All single-cell experiments were performed by R.S. and T.V.B., with help from S.C. Data analysis was performed by S.C. and G.P. with substantial contribution from B.B. The manuscript was written by S.C. under the supervision of D.L. and J.D. I.A., A.M., A.B., F.L., O.B., J.Q. contributed critical data interpretation. All authors read or provided comments on the manuscript.

## Competing interests

The authors declare no competing interests.

## Additional information

[1]Digestive Oncology, Department of Gastroenterology, University Hospitals Leuven, Leuven, Belgium. [2]Laboratory of Clinical Digestive Oncology, Department of Oncology, KU Leuven, Leuven, Belgium. [3]Laboratory for Translational Genetics, Department of Human Genetics, KU Leuven, Leuven, Belgium. [4]VIB Centre for Cancer Biology, Leuven, Belgium. [5]Radiology Department, University Hospitals Leuven, Leuven, Belgium. [6]Laboratory of Translational MRI, Department of Imaging and Pathology, KU Leuven, Leuven, Belgium. [7]Hepatobiliary- and pancreas Surgery, Department of Abdominal Surgery, University Hospitals Leuven, Leuven, Belgium. [8]Zhejiang Provincial Key Laboratory of Precision Diagnosis and Therapy for Major Gynaecological Diseases, Women's Hospital, Zhejiang University School of Medicine, Hangzhou, China. [9]Institute of Genetics, Zhejiang University School of Medicine, Hangzhou, China. [10]These authors contributed equally: Diether Lambrechts, Jeroen Dekervel. ✉e-mail: diether.lambrechts@kuleuven.be; jeroen.dekervel@uzleuven.be

