## [Peer review file · Nature Communications]

REVIEWER COMMENTS

Reviewer #1 (expert in gastroenterology and hepatocellular carcinoma):

Thank you very much for inviting me to review PD-1 negative CD45RA effector-memory CD8 T-cells and CXCL10 macrophages are 1 essential for response to checkpoint inhibition in advanced hepatocellular carcinoma by Cappuyns et al. In this paper, authors study the role of PD1-expressing CD8 T-cells and their role in response to immunotherapy. Through single-cell profiling, authros found that advanced HCC tumors with durable response to checkpoint blockade had an abundance of PDL1-expressing CXCL10+ macrophages and high levels of CXCL9/10/11, attracting CXCR3+ effector-memory T-cells (CD8 TEM) into the tumor. These CD8 TEM cells preferentially differentiated into clonally-expanded, PD1-negative, CD45RA effector-memory CD8 T-cells (CD8 TEMRA) with strong cytotoxic activity, highlighting a unique mechanism of response to immunotherapy in advanced HCC.

The study addresses an important question in HCC research. Has important data and is written well. However, the small number of samples, heterogeneous nature of the cohort and strong conclusions not fully supported by their data, dampen my enthusiasm. Here are some major concerns-

Concerns about cohort-

A significant proportion of patients have received various types of therapies prior to biopsy which can serve as a major confounding factor for how the tumor microenvironment was remodeled. The authors start with "homogenous single-cell atlas of both the TME and peripheral immune". However, this still is a small study and not all patients have PBMCs from the pre-designed time points. Even though authors simplify it as "(week 0-3-6) PBMC samples (n=58)", this complete data was only available for around 13 patients who received ICI. The treatment received by this cohort is also not homogenous. Patients received either monotherapy with ICI or combination with TKI.

Figure 1 and 2 are both descriptive, mostly profiling the various types of cells identified, a lot of this can be moved to supplementary.

The main result i.e abundance of CD8 Termra cells in responders is only marginally significant with wide variation in distribution noted in Fig 3A. Given the small number of patients, it looks like the outliers might have a big influence.

Throughout the manuscript, it is not clear if the progression-free survival is adjusted for key clinical variables. And also why not show Overall survival? Moreover, the number of events in both Fig 3D, 4B, 6B, 7 are so small that survival analysis is difficult to interpret and caution needs to be exerted.

Despite this being a correlative study, authors use a lot of strong language suggesting causation like- "PD-1 negative CD45RA effector-1 memory CD8 T-cells and CXCL10 macrophages are essential for response to checkpoint inhibition"

"pivotal role in facilitating response to CPI."

"PD1-expressing CD8 T-cells do not become activated during response to CPI"

"confirming their value as predictive biomarkers response to CPI in aHCC"

These significant claims are not entirely supported by their data and are especially challenging given the small number of samples.

Reviewer #2 (expert in hepatocellular carcinoma, biomarkers of drug resistance):

This is a well-written paper studying a cohort of patients with HCC treated with IOs where sc RNA seq was used to identify CXCL9/10/11 and CD45RA effector-memory CD8 T-cells as potential mediators of

treatment outcome.

This study provides a nice resource of scRNA-seq data, but the analyses of peripheral blood seems entirely superfluous, though it does provide confirmatory evidence that tumor reactive immune cells circulate in HCC. Given the large amount of data in this paper, I would recommend that these analyses be abbreviated, most of it is already in the supplemental. The validation using IMbrave150 data is fantastic.

The major issue I have is that this study, just like all other descriptive omic papers, show no causation. This is not a flaw of this study, but the language used throughout the paper confuses this. For instance, in the 2nd sentence, the authors state: "CD45RA effector-memory CD8 T-cells (CD8 TEMRA) play a pivotal role in facilitating response to CPI". Other sentences in the Results and Discussion are affected.

Overall, great work!

Reviewer #3 (expert in computational biology, scRNAseq, and TCRseq):

This manuscript presents an important finding on the mechanisms underlying HCC patients responding to immunotherapy. Overall, this study is well designed, well conducted, and well organized. The usage of scRNA-seq and TCR-seq is elegant, with proper integration of the information presented by each technology. The discovery is also consistent with many published papers, which indicate the finding of this manuscript but do not demonstrate it because of the absence of participants receiving immunotherapy. Overall, the finding of this manuscript is exciting, stimulating, and promising. I only have one minor question for the authors to address before acceptance for publication.

Minor concern:

In the current manuscript, the authors depicted the macrophage subset critical for immunotherapy responses as PDL1 -expression and CXCL10+. CXCL10 is important to recruit peripheral Temra. My concern is whether PDL1 expression is also the same important. I suggest the authors conduct a co-expression analysis of PDL1 and CXCL10 within the whole macrophages and this PDL1-expression CXCL10+ macrophage subset. The results will be helpful for readers to understand the critical phenotypes of this important macrophage subset.

For data sharing, I suggest the authors to upload an expression matrix and the corresponding metadata to the NCBI GEO database, which is well-maintained and will further amplify the usage of the data.

Reviewer #1 (expert in gastroenterology and hepatocellular carcinoma):

Thank you very much for inviting me to review PD-1 negative CD45RA effector-memory CD8 T-cells and CXCL10 macrophages are essential for response to checkpoint inhibition in advanced hepatocellular carcinoma by Cappuyns et al. In this paper, authors study the role of PD1-expressing CD8 T-cells and their role in response to immunotherapy. Through single-cell profiling, authors found that advanced HCC tumors with durable response to checkpoint blockade had an abundance of PDL1-expressing CXCL10+ macrophages and high levels of CXCL9/10/11, attracting CXCR3+ effector-memory T-cells (CD8 TEM) into the tumor. These CD8 TEM cells preferentially differentiated into clonally-expanded, PD1-negative, CD45RA effector-memory CD8 T-cells (CD8 TEMRA) with strong cytotoxic activity, highlighting a unique mechanism of response to immunotherapy in advanced HCC.

The study addresses an important question in HCC research. Has important data and is written well. However, the small number of samples, heterogeneous nature of the cohort and strong conclusions not fully supported by their data, dampen my enthusiasm. Here are some major concerns-

Comment #1:

Concerns about cohort:

A significant proportion of patients have received various types of therapies prior to biopsy which can serve as a major confounding factor for how the tumor microenvironment was remodeled. The authors start with “homogenous single-cell atlas of both the TME and peripheral immune”. However, this still is a small study and not all patients have PBMCs from the pre-designed time points. Even though authors simplify it as “(week 0-3-6) PBMC samples (n=58)”, this complete data was only available for around 13 patients who received ICI.

The treatment received by this cohort is also not homogenous. Patients received either monotherapy with ICI or combination with TKI.

We want to thank Reviewer #1 for his comments. He raises **two** very valid issues concerning the single-cell cohort, namely that i) the cohort is relatively small and ii) there is a considerable degree of heterogeneity within the treatment groups. In this revised manuscript, we have aimed to address both comments.

Firstly, we have significantly increased the size of the cohort to include a total of **44 advanced HCC patients**. The number of tumour biopsies was increased to include a total of 38 pre-treatment tumour biopsies (instead of 31) and additional blood samples were included for a total of 72 PBMC samples (instead of 58). The additional samples have also ensured that the size of the ‘overlap dataset’ (*i.e.* patients with both tumour biopsy and PBMC samples available) was increased to include a total of 19 patients (instead of 13). For details of the sample availability per patient please refer to **Supplementary Table S2** in the revised manuscript). All analyses and figures in the main manuscript and supplementary files have been redone, integrating the additional samples.

Importantly, we would like to underline the fact that single-cell data in HCC is scarce. This is related to the paucity of tumour tissue available for translational research, in a disease where diagnostic tumour biopsies are not mandated by current clinical guidelines. Consequently, translational research most often relies on (often archival) tumour resection specimens from early HCC. **Table R1** provides an overview of all HCC single-cell datasets published so far. Nine (out of only 11 studies) were conducted in early HCC, including tumour resections and therefore HCC patients that did not receive systemic therapy. These cohorts do not allow the correlation of single-cell readouts with response to systemic therapy. Only one other cohort^{1,2} included HCC patients treated with systemic therapy (n=18), though the cohort was also heterogenous in both tumour type (including both HCC and intrahepatic cholangiocarcinoma) and tumour stage (early versus advanced stages).

Our updated cohort now represents the **largest** single-cell cohort that focusses on **advanced HCC** patients (**n=44**) treated with systemic therapy that allows the correlation of single-cell readouts with response to therapy.

Secondly, the heterogeneity in treatment groups is certainly an important limitation. This is related to the fast pace at which the treatment landscape of advanced HCC has evolved since the start of our study. In order to ensure enough power to identify rare cell populations, all samples (irrespective of treatment administered) were used to annotate single cells into their respective cell (pheno-)types. However, to address the treatment heterogeneity, we have chosen to focus all analyses comparing responders to non-responders on those patients treated with **atezolizumab plus bevacizumab** (atezo/bev). We believe that focussing on atezo/bev, current standard of care in first line treatment of advanced HCC is a **more relevant** question in the HCC treatment landscape today. Firstly, because so far there are no papers that correlate single-cell readouts with response to atezo/bev (**Table R1**), and secondly as atezo/bev is increasingly being considered in earlier lines of treatment (IMBrave050³), biomarkers of response to atezo/bev are of great interest to our scientific community.

Finally, Reviewer #1 correctly mentions that a proportion of patients (15 out of 44) underwent previous treatments for HCC (see **Supplementary Table S1** and **Table R2** for details). As single-cell sequencing requires fresh tissue dissociation, we cannot rely on archival tumour tissue and all patients included in our study underwent a tissue biopsy at the time of inclusion, prior to starting atezo/bev. Similarly, the four patients previously exposed to systemic therapy (tyrosine kinase inhibitors or chemotherapy), underwent a new tissue biopsy prior to starting immunotherapy-based regimens. Importantly, none of the included patients had been exposed to immunotherapy prior to enrolment. Nevertheless, prior treatments can certainly alter or remodel the tumour-microenvironment and therefore influence subsequent response to immunotherapy-based regimens. **Table R2** provides per-patient data on the 15 previously treated patients. Considering all comparative analyses in the revised manuscript are focused on atezo/bev treated patients, only 6 patients (3 responders and 3 non-responders) were previously treated with locoregional therapies, liver resection or a combination of both. None were pre-treated with systemic therapies. Considering this small number of pre-treated patients, we believe the role of previous HCC treatment as a confounding factor in this cohort is limited. Most importantly, in the **larger** and more **homogenous** cohort of atezo/bev treated patients included in the revised manuscript, we were able to replicate all our findings, underlining their robustness.

Table R1 : Overview of HCC datasets using single-cell sequencing.

Reference	Patients (n)	Tumour type	Systemic Treatment	Sample type	Technology
Zheng et al. 2017 ⁴ and Zhang et al. 2019 ⁵	15	Early HCC	None	Tumour Adjacent Liver Lymph node PBMC, ascites	scRNAseq
Losic et al. 2020 ⁶	2	Early HCC	None	Tumour	scRNAseq
Ho et al. 2021 ⁷	8	Early HCC	None	Tumour	scRNAseq
Song et al. 2020 ⁸	7	Early HCC	None	Tumour	scRNAseq
Lu et al. 2022 ⁹	9	Early HCC	None	Tumour Adjacent Liver	scRNAseq
Sun et al. 2021 ¹⁰	18	Early HCC	None	Tumour Adjacent Liver	scRNAseq
Xue et al. 2023 ¹¹	79	Early HCC	None	Tumour Adjacent Liver	scRNAseq
Ma et al. 2022 ¹²	4	Early HCC	None	Tumour Tumour border Adjacent liver	scRNAseq scTCRseq
Sharma et al. 2020 ¹³	13	Early HCC	None	Tumour	scRNAseq
Ma et al. 2019 and 2021 ^{1,2}	25	21 advanced HCC 4 early HCC	18 durva/treme 7 None	Tumour	scRNAseq
Magen et al. 2023 ¹⁴	20	Early HCC	20 anti-PD1	Tumour (post-treatment)	scRNAseq scTCRseq
Cappuyns et al.	44	Advanced HCC	25 atezo/bev 1 atezo/cabo 11 anti-PD(L)1 5 TKI 2 None	Tumour PBMC	scRNAseq scTCRseq

Table R2. Overview of patients previously treated for HCC

Patient ID	Biopsy Available	Previous Treatment	Treatment	Response
HCC003	yes	Resection; TKI	Anti-PD1	NonResponder
HCC006	yes	TKI	Anti-PD1	DeathBeforeImaging
HCC008	yes	Resection; Chemotherapy; TKI	None	NoTreatment
HCC009	yes	Local	Anti-PD1	Responder
HCC010	yes	Local	TKI	Responder
HCC016	yes	Resection; Local	Atezo/bev	NonResponder
HCC017	yes	Resection	Atezo/bev	Responder
HCC026	yes	Local	Atezo/bev	Responder
HCC028	yes	Local	Atezo/bev	NonResponder
HCC030	yes	Local	Atezo/bev	NonResponder
HCC032	yes	Liver transplantation	TKI	NonResponder
HCC035	yes	Liver transplantation	TKI	NonResponder
HCC046	yes	Local	Atezo/bev	Responder
HCCX4	no	Chemotherapy	Atezo/bev	NonResponder
HCCX7	no	Local	Atezo/bev	Responder

Abbreviations: TKI = tyrosine kinase inhibitor; atezo/bev = atezolizumab plus bevacizumab; atezo/cabo = atezolizumab plus cabozantinib; PD1 = Programmed Cell Death Protein 1.

Comment #2:

Figure 1 and 2 are both descriptive, mostly profiling the various types of cells identified, a lot of this can be moved to supplementary.

We agree that Figure 1 and 2 are mostly descriptive, and take away from the main message of the manuscript, namely understanding response to atezo/bev in advanced HCC. Therefore, we shortened the description in the text, combined Figure 1 and 2 (now **Figure 1 in revised manuscript**) and the remaining figures were moved to the Supplementary Figures (**Supplementary Figure 1 and 2 in revised manuscript**).

Comment #3:

The main result i.e abundance of CD8 Termra cells in responders is only marginally significant with wide variation in distribution noted in Fig 3A. Given the small number of patients, it looks like the outliers might have a big influence.

The association of CD8 T_{EMRA} with response to atezo/bev is indeed one of the main findings in our study. However, we would like to emphasize that this finding is not only based on cell abundancies, but on **three independent observations**, outlined below:

1. CD8 T_{EMRA} : relative abundancies

In the **discovery** cohort, the relative abundance of CD8 T_{EMRA} is significantly associated with response to atezo/bev, both when corrected for the total number of intra-tumoural T-cells per sample (p=0.04; **Fig. 2e in revised manuscript**) and the total number of intra-tumoural CD8 T-cells (p=0.049; **Supplementary Fig. 3c in revised manuscript**). Undoubtedly, the relative abundance of cell types in single-cell RNA sequencing data is strongly influenced by the sampling bias that is inherent to single-cell technologies. This is even more relevant for rare cell populations, such as CD8 T_{EMRA} that represent <5% of all intra-tumoural T-/NK-cells. Consequently, 'relative abundance' is one of the least robust single-cell readouts mentioned in the manuscript, resulting in the large variation in values. Nevertheless, based on differential gene expression analysis, responding tumours overexpress typical CD8 T_{EMRA} marker genes (*FGFBP2*, *GPLY*, *SPON2*; **Fig. 2f in revised manuscript**).

Additionally, we used MiloR¹⁵, a tool for differential abundance testing in single-cell data that applies k-nearest neighbour graphs and therefore less susceptible to outliers or individual patients. The analysis confirmed the association of CD8 Temra with response to atezo/bev.

2. CD8 T_{EMRA} : T-cell expansion and TCR clonality

Independent from the differences in cell abundancies, we observed that CD8 T_{EMRA} from responders specifically are more clonally-expanded than those derived from non-responding tumours (**Fig. 2h right in revised manuscript**), while there are no significant differences in clonal expansion within the CD8 T_{EX} and CD8 T_{EM} according to clinical response.

3. CD8 T_{EMRA} : TCR sharing with peripheral blood

Finally, we consider the analysis exploring TCRs found both in the tumour and in peripheral blood prior to treatment, so-called TCR sharing, one of the most convincing findings:

- a) Atezo/bev responders display a significantly higher degree of TCR sharing compared to non-responders (**Fig. 3a-b; Supplementary Fig. 3g in revised manuscript**). This findings supports the hypothesis that intra-tumoural T-cells with TCRs also found in blood prior to treatment, could represent baseline anti-tumour immunity.
- b) These 'shared TCRs' are concentrated within intra-tumoural CD8 T_{EMRA} from responders, specifically (**Fig. 3c in revised manuscript**).
- c) Peripheral CD8 Temra also displayed the highest degree of TCR sharing, a phenomenon almost exclusively observed in responders which persisted on treatment, supporting the hypothesis that CD8 T_{EMRA} carry TCRs that target tumour antigens essential to achieve durable response to atezo/bev (**Fig. 4g. in revised manuscript**).

Based on these multiple lines of evidence, we are convinced that our findings pointing towards CD8 T_{EMRA} as important cell phenotypes in response to atezo/bev in advanced HCC are robust.

Importantly, we consider the single-cell cohort a **discovery** cohort that is well-suited for in-depth analysis of a limited number of samples. We then confirm the two main findings from our study in a large **validation cohort** of bulk RNA sequencing data¹⁶. A total of 311 prospectively collected advanced HCC tumours treated with atezo/bev (n=253) versus sorafenib (n=58) were used to demonstrate that the presence of Macro CXCL10 and CD8 T_{EMRA} prior to treatment (as defined by a combined gene signature score) is associated with improved overall *and* progression free survival upon treatment with atezo/bev. An association that was *not* seen in the, albeit relatively small (n=58), subset of patients treated with sorafenib.

Comment #4:

Throughout the manuscript, it is not clear if the progression-free survival is adjusted for key clinical variables. And also why not show Overall survival? Moreover, the number of events in both Fig 3D, 4B, 6B, 7 are so small that survival analysis is difficult to interpret and caution needs to be exerted.

In the single-cell **discovery** cohort, the primary aim of our study was to identify factors associated with response versus resistance to atezo/bev. Patients were therefore stratified based on clinical response, which was considered the primary endpoint of our study. Several parameters associated with response were then also tested for association with progression free survival (PFS), as a secondary endpoint. However, considering the size of the discovery cohort and the limited number of events, it was not statistically feasible to correct for baseline patient or tumour characteristics in a multivariate analysis. We fully agree that the analyses must be interpreted accordingly. We have added a statement in the Methods section to make this clear (see lines 693-695 in revised manuscript).

Additionally, we did not consider overall survival as a secondary endpoint, as this parameter is strongly influenced by other factors such as unrelated causes of death, particularly in a fragile HCC patient population, or as the treatment options for advanced HCC continues to expand, many patients are eligible for second line systemic therapies upon progression under atezo/bev. In our cohort, 6 (out of 44) patients died from unrelated causes of deaths and out of the 25 atezo/bev treated patients, 13 received received tyrosine kinase inhibitors (n=11) or chemotherapy (n=2) upon progression with atezo/bev. Given these numbers, we did not consider overall survival in the **discovery** phase of our study.

The **validation** cohort is a publicly available, prospectively collected cohort¹⁶ of patients treated with atezo/bev (n=253) or sorafenib (n=58) in the context of the phase Ib (GO30140¹⁷; arms A and F) and phase III (IMBrave150^{18,19}) clinical trials that established atezo/bev as standard of care in first line treatment of advanced HCC. In the validation cohort we demonstrate that the presence of CD8 T_{EMRA} and Macro CXCL10 in the tumour-microenvironment prior to treatment (as defined by a combined gene signature score) is associated with longer overall *and* progression free survival. Unfortunately, these data are publicly available and additional patient or tumour characteristics were not available to perform a multivariate cox regression analysis. We have added a sentence in the limitation section to acknowledge this (see lines 448-450 in revised manuscript) as well as a statement in the Methods section (see lines 732-733 in revised manuscript).

Comment #5:

Despite this being a correlative study, authors use a lot of strong language suggesting causation like-
“PD-1 negative CD45RA effector-1 memory CD8 T-cells and CXCL10 macrophages are essential for response to checkpoint inhibition”

“pivotal role in facilitating response to CPI.”

“PD1-expressing CD8 T-cells do not become activated during response to CPI”

“confirming their value as predictive biomarkers response to CPI in aHCC”

These significant claims are not entirely supported by their data and are especially challenging given the small number of samples.

We want to thank Reviewer #1 for this comment, which is in line with comment #1 from Reviewer #2. Descriptive single-cell sequencing is ideal for in depth analysis of a limited number of samples and are therefore appropriate to explore associations. The single-cell cohort is indeed a **discovery** cohort that allows us to derive several factors *associated with* response to atezo/bev. The *association* of the two putative biomarkers with clinical outcome (overall and progression free survival) upon treatment with atezo/bev was also **validated** using a large, prospectively collected bulk RNA-sequencing cohort¹⁶, consisting of pre-treatment tumour biopsies of advanced HCC patients treated with atezo/bev (n=253) or sorafenib (n=58). However, we fully agree that we cannot demonstrate causation and the language used may sometimes confuse this point. Therefore, we have adapted the phrasing where necessary (for example: see lines 97-98, 168, 178-179, 268, 277, 339-441, 344-345, 368-369, 404, 418, 420-422, 432-433 in the revised manuscript). We have also addressed this as a limitation in the discussion of the revised manuscript (lines 452-453).

Nevertheless, we would like to reiterate the fact that several independent factors point towards the CD8 T_{EMRA} as the potential effector cells in response to atezo/bev. Together with the on-treatment dynamics of CD8 T_{EMRA} in peripheral blood, these findings support the hypothesis that CD8 T_{EMRA} are potential mediators of response atezo/bev in advanced HCC, and not just bystander T-cells. Regarding the CXCL10+ macrophages, we confirm previous findings by other groups²⁰.

Reviewer #2 (expert in hepatocellular carcinoma, biomarkers of drug resistance):

This is a well-written paper studying a cohort of patients with HCC treated with IOs where sc RNA seq was used to identify CXCL9/10/11 and CD45RA effector-memory CD8 T-cells as potential mediators of treatment outcome.

Comment #1:

This study provides a nice resource of scRNA-seq data, but the analyses of peripheral blood seems entirely superfluous, though it does provide confirmatory evidence that tumor reactive immune cells circulate in HCC. Given the large amount of data in this paper, I would recommend that these analyses be abbreviated, most of it is already in the supplemental. The validation using IMbrave150 data is fantastic.

Figure 1 and 2 are mostly descriptive, elaborating on the annotation of the various intra-tumoural and peripheral cell (pheno-)types. While we believe the annotation of the intra-tumoural cell types to be very relevant, considering our cohort is the first dataset to explore the tumour-microenvironment of HCC patients at advanced stages of the disease, we do agree that including extensive details on the annotation of the PBMCs is repetitive, taking away from the main message of the manuscript, namely understanding response to atezo/bev in advanced HCC. Therefore, we have shortened the description in the text, combined Figure 1 and 2 (now **Figure 1 in revised manuscript**) and moved the remaining figures related to cell (pheno-)type annotation to the Supplementary Figures (**Supplementary Figure 1 and 2 in revised manuscript**).

However, we would like to point out, that we consider the analysis exploring TCRs found both in the tumour and in peripheral blood prior to treatment particularly interesting (**Fig. 3 and Fig. 4 in the revised manuscript**) and the PBMC dataset is an essential part of these findings. To improve the robustness of our findings, we have increased the number of tumour (n=38 instead of 31) and PBMC samples (n=72 instead of 58) in the revised manuscript. With the additional samples, the 'overlap dataset', *i.e.* patients with both tumour biopsy and PBMC samples available, now includes a total of 19 patients (instead of 13).

Furthermore, we have also opted to focus the comparative analysis on those patients treated with atezo/bev. Importantly, in the larger and more homogenous cohort included in the revised manuscript, our findings were confirmed. Not only do atezo/bev responders display a significantly higher degree of TCR sharing compared to non-responders (**Fig. 3a-b; Supplementary Fig. 3g in revised manuscript**), these 'shared TCRs' are concentrated within intra-tumoural CD8 T_{EMRA} from responders, specifically (**Fig. 3c in revised manuscript**). Importantly, this TCR sharing in CD8 T_{EMRA} persisted on treatment in responders. This was in clear contrast with the TCRs found in intra-tumoural CD8 T_{EX} that were present in <1% of peripheral CD8 T-cells in responders and non-responders alike; nor did these TCRs emerge in peripheral blood during treatment (**Fig. 4g. in revised manuscript**). These data support the potential role of CD8 T_{EMRA} as effector cells upon response to atezo/bev in advanced HCC, a truly unique concept that differs from data in most other cancer types that we certainly wish to highlight in the manuscript.

Comment #2:

The major issue I have is that this study, just like all other descriptive omic papers, show no causation. This is not a flaw of this study, but the language used throughout the paper confuses this. For instance, in the 2nd sentence, the authors state: "CD45RA effector-memory CD8 T-cells (CD8 TEMRA) play a pivotal role in facilitating response to CPI". Other sentences in the Results and Discussion are affected.

This comment is in line with comment #5 From Reviewer #1. Indeed, while single-cell sequencing is ideal for in-depth analysis of a limited number of samples, these studies are descriptive and therefore only suitable for associations. Although, the *association* of the two putative biomarkers with clinical outcome (overall and progression free survival) upon treatment with atezo/bev was also **validated** using a large, prospectively collected bulk RNA-sequencing cohort¹⁶, consisting of pre-treatment tumour biopsies of advanced HCC patients treated with atezo/bev (n=253) or sorafenib (n=58). However, we fully agree that we cannot demonstrate causation and the language used may sometimes confuse this point. Therefore, we have adapted the phrasing where necessary (for example: see lines

97-98, 168, 178-179, 268, 277, 339-441, 344-345, 368-369, 404, 418, 420-422, 432-433 in the revised manuscript). We have also addressed this as a limitation in the discussion of the revised manuscript (lines 452-453).

Nevertheless, we would like to reiterate the fact that several independent factors point towards the CD8 T_{EMRA} as the potential effector cells in response to atezo/bev. Together with the on-treatment dynamics of CD8 T_{EMRA} in peripheral blood, these findings support the hypothesis that CD8 T_{EMRA} are mediators of response atezo/bev in advanced HCC, and not just bystander T-cells. Regarding the CXCL10+ macrophages, we confirm previous findings by other groups²⁰.

Overall, great work!

We would like to thank Reviewer #2 for these kind words and positive feedback on our work!

Reviewer #3 (expert in computational biology, scRNAseq, and TCRseq):

This manuscript presents an important finding on the mechanisms underlying HCC patients responding to immunotherapy. Overall, this study is well designed, well conducted, and well organized. The usage of scRNA-seq and TCR-seq is elegant, with proper integration of the information presented by each technology. The discovery is also consistent with many published papers, which indicate the finding of this manuscript but do not demonstrate it because of the absence of participants receiving immunotherapy. Overall, the finding of this manuscript is exciting, stimulating, and promising. I only have one minor question for the authors to address before acceptance for publication.

We thank Reviewer #3 for these kind words and obvious enthusiasm for our work!

Comment #1:

Minor concern:

In the current manuscript, the authors depicted the macrophage subset critical for immunotherapy responses as PDL1 -expression and CXCL10+. CXCL10 is important to recruit peripheral Temra. My concern is whether PDL1 expression is also the same important. I suggest the authors conduct a co-expression analysis of PDL1 and CXCL10 within the whole macrophages and this PDL1-expression CXCL10+ macrophage subset. The results will be helpful for readers to understand the critical phenotypes of this important macrophage subset.

CXCL10+ macrophages are annotated based on their expression of known marker genes including (*CXCL9, CXCL10, CXCL11, STAT1, GBP5*; **Supplementary Fig. 5e-f in revised manuscript**). They are a very small subset of macrophages present within the TME that represent <1% of all cells identified. On the other hand, PDL1 (*CD274*) in the TME was generally low (**Supplementary Fig. 5a-b in revised manuscript**) but detectable within the myeloid cells.

As suggested, we performed a co-expression analysis in all monocytes and macrophages in the TME. Within CXCL10+ macrophages (Macro CXCL10), PDL1 was expressed in approximately 20% of cells,

while in all other myeloid subtypes, PDL1 was expressed in just 3% of cells, on average (range 0.1%-7.4%). Furthermore, within PDL1-expressing CXCL10+ macrophage, 85% of cells also expressed CXCL10, and mostly in those CXCL10+ macrophages from responders (Fig. 5c bottom; Supplementary Fig. 6c in revised manuscript). To illustrate this further, Fig. R1 depicts the expression of CXCL10 versus CD274 expression and the combination of both in the intra-tumoural myeloid compartment. The figure was added to the revised manuscript as Supplementary Fig. 6d.

Figure R1 | Co-expression analysis CD274 (PDL1) and CXCL10. *Left:* UMAP representation of intra-tumoural monocytes and macrophages. *Right:* UMAP representation depicting expression levels of CXCL10, CD274 and the combination of both.

Comment #2:

For data sharing, I suggest the authors to upload an expression matrix and the corresponding metadata to the NCBI GEO database, which is well-maintained and will further amplify the usage of the data.

We thank Reviewer #3 for this suggestion. We have opted to upload the raw sequencing reads of the scRNAseq, scTCRseq and Totalseq-C experiments to the European Genome-phenome Archive (EGAS00001007547) database. In accordance with the European GDPR law, requests for accessing raw sequencing reads will be reviewed by the UZLeuven-VIB data access committee. Any data shared will be released via a Data Transfer Agreement that will include the necessary conditions to guarantee protection of personal data. Alternatively, a download of the read count data per sample, necessary to reproduce all analyses included in this manuscript, will be made available at <https://lambrechtslab.sites.vib.be/en/data-access>. Several published datasets from our group are available here, and the website has a significant outreach with >1000 data downloads per year.

Also, to ensure reproducibility, we have uploaded a Source Data file, which includes a separate sheet per figure, containing the raw data underlying each figure. A statement referring to the availability of the Source Data was added to the Data Availability statement.

References

1. Ma, L., Hernandez, M. O., Zhao, Y. *et al.* Tumor Cell Biodiversity Drives Microenvironmental Reprogramming in Liver Cancer. *Cancer Cell* **36**, 418–430.e6. (2019).
2. Ma, L., Wang, L., Khatib, S. A. *et al.* Single-cell atlas of tumor cell evolution in response to therapy in hepatocellular carcinoma and intrahepatic cholangiocarcinoma. *J Hepatol* **75**, 1397–1408 (2021).

3. Chow, P., Chen, M., Cheng, A. *et al.* IMbrave050: Phase 3 study of adjuvant atezolizumab + bevacizumab versus active surveillance in patients with hepatocellular carcinoma (HCC) at high risk of disease recurrence following resection or ablation [Abstract CT003]. *Cancer Res.* **83** (2023).
4. Zheng, C., Zheng, L., Yoo, J. *et al.* Landscape of Infiltrating T Cells in Liver Cancer Revealed by Single-Cell Sequencing. *Cell* **169**, 1342-1356.e16 (2017).
5. Zhang, Q., He, Y., Luo, N. *et al.* Landscape and Dynamics of Single Immune Cells in Hepatocellular Carcinoma. *Cell* **179**, 829-845.e20 (2019).
6. Losic, B., Craig, A. J., Villacorta-Martin, C. *et al.* Intratumoral heterogeneity and clonal evolution in liver cancer. *Nat Commun* **11**, 291 (2020).
7. Ho, D. W.H., Tsui, Y.M., Chan, L.K. *et al.* Single-cell RNA sequencing shows the immunosuppressive landscape and tumor heterogeneity of HBV-associated hepatocellular carcinoma. *Nat Commun* **12**, 3684 (2021).
8. Song, G., Shi, Y., Zhang, M. *et al.* Global immune characterization of HBV/HCV-related hepatocellular carcinoma identifies macrophage and T-cell subsets associated with disease progression. *Cell Discov* **6**, 90 (2020).
9. Lu, Y., Yang, A., Quan, C. *et al.* A single-cell atlas of the multicellular ecosystem of primary and metastatic hepatocellular carcinoma. *Nat Commun* **13**, 4594 (2022).
10. Sun, Y., Wu, L., Zhong, Y. *et al.* Single-cell landscape of the ecosystem in early-relapse hepatocellular carcinoma. *Cell* **184**, 404-421.e16 (2021).
11. Xue, R., Zhang, Q., Cao, Q. *et al.* Liver tumour immune microenvironment subtypes and neutrophil heterogeneity. *Nature* **612**, 141-147 (2022).
12. Ma, L., Heinrich, S., Wang, L. *et al.* Multiregional single-cell dissection of tumor and immune cells reveals stable lock-and-key features in liver cancer. *Nat Commun* **13**, 7533 (2022).
13. Sharma, A., Seow J. J. W., Dutertre, C. A. *et al.* Onco-fetal Reprogramming of Endothelial Cells Drives Immunosuppressive Macrophages in Hepatocellular Carcinoma. *Cell* **183**, 377-394.e21 (2020).
14. Magen, A., Hamon, P., Fiaschi, N. *et al.* Intratumoral dendritic cell – CD4 + T helper cell niches enable CD8 + T cell differentiation following PD-1 blockade in hepatocellular carcinoma. *Nat Med* **29**, 1389-1399 (2023).
15. Dann, E., Henderson, N. C., Teichmann, S. A. *et al.* Differential abundance testing on single-cell data using k-nearest neighbor graphs. *Nat Biotechnol* **40**, 245–253 (2022).
16. Zhu, A. X., Abbas, A. R., Ruiz de Galarreta, M. *et al.* Molecular correlates of clinical response and resistance to atezolizumab in combination with bevacizumab in advanced hepatocellular carcinoma. *Nat Med* **28**, 1599–1611 (2022).
17. Lee, M., Ryoo, B., Hsu, C. *et al.* Atezolizumab with or without bevacizumab in unresectable hepatocellular carcinoma (GO30140) : an open-label, multicentre phase 1B study. *Lancet Oncol.*

- 21, 808–820 (2020).
18. Finn, R. S., Qin, S., Ikeda, M. *et al.* Atezolizumab Plus Bevacizumab in Unresectable Hepatocellular Carcinoma. *N Engl J Med* **382**, 1894–1905 (2020).
 19. Cheng, A. L., Qin, S., Ikeda, M. *et al.* Updated efficacy and safety data from IMbrave150: Atezolizumab plus bevacizumab vs. sorafenib for unresectable hepatocellular carcinoma. *J Hepatol* **76**, 862–873 (2022).
 20. Kaya, N. A., Tai, D., Lim, X. *et al.* Multimodal molecular landscape of response to Y90-resin microsphere radioembolization followed by nivolumab for advanced hepatocellular carcinoma. *J Immunother Cancer* **11**, e007106 (2023).

REVIEWERS' COMMENTS

Reviewer #1 (expert in gastroenterology and hepatocellular carcinoma):

Thank you very much for inviting me to review the revised version of the manuscript PD-1 negative CD45RA effector-memory CD8 T-cells and CXCL10 macrophages underlie response to atezolizumab/bevacizumab in advanced hepatocellular carcinoma by Cappuyns et al. In this paper, authors study the role of PD1-negative CD8 T-cells and their role in response to Atezo/Bev. Through single-cell profiling, authors found that advanced HCC tumors with durable response to checkpoint blockade had an abundance of PDL1-expressing CXCL10+ macrophages and high levels of CXCL9/10/11, potentially attracting CXCR3+ effector-memory T-cells (CD8 TEM) into the tumor. These CD8 TEM cells preferentially differentiated into clonally-expanded, PD1-negative, CD45RA effector-memory CD8 T-cells (CD8 TEMRA) with strong cytotoxic activity, highlighting a unique mechanism of response to immunotherapy in advanced HCC.

Authors have addressed my concerns and expanded the number of samples. Some of the language has also been modified. However, I have a few suggestions to further clarify the statements in the manuscript. Would also suggest improvements to the figure legends.

Title says "PD-1 negative CD45RA effector-memory CD8 T-cells and CXCL10 macrophages 1 underlie response to atezolizumab/bevacizumab in advanced hepatocellular carcinoma ". This still suggests a causal link between these immune subsets and response to Atezo/Bev. Would suggest change to "PD-1 negative CD45RA effector-memory CD8 T-cells and CXCL10 macrophages are associated with response to atezolizumab/bevacizumab in advanced hepatocellular carcinoma ".

In the abstract they say "Tumours with durable response were enriched for PDL1- expressing CXCL10+ macrophages and, based on cell-cell interaction, expressed high levels of CXCL9/10/11 to attract peripheral CXCR3+ effector-memory T-cells (CD8 TEM) into the tumour." This suggests a functional link. Would stick to correlative language. "Tumours with durable response were enriched for PDL1- expressing CXCL10+ macrophages which also express high levels of CXCL9/10/11 and potentially recruit peripheral CXCR3+ effector-memory T-cells (CD8 TEM) into the tumour."

Please clarify in figure legends, not clear from the figure legends-

Fig 2a-2d- UMAP of which samples ?n

Fig 2e and 2g says Atezo-Bev n=20. Figure 1 said this group was 25?. How many in this graph are responders vs.nonresponders

Fig 2f How many T cells in responders and nonresponders?

Fig 3a-b, how many responders and non-responders

Figure 4- how many T cells?

Figure 5b, how many samples, how many responders and non-responders

Fig 5c- why are there 20 patients to show PDL1 expression but only 17 for macro CXCL10

Reviewer #2 (expert in hepatocellular carcinoma, biomarkers of drug resistance):

The authors have done a substantial amount of work in this revision, which is fantastic. I have only a minor concern about the use of "unique" in the last sentence of the abstract, as this study does not really offer a direct comparison of atezo bev versus other therapy outcomes.

Reviewer #3 (expert in computational biology, scRNAseq, and TCRseq):

The authors have almost addressed all my questions except that the gene expression matrix is maintained at the authors' own website instead of being deposited to a public database. Although it is

not ideal, it is acceptable. Overall, it is a great work.

Reviewer #1 (expert in gastroenterology and hepatocellular carcinoma):

Thank you very much for inviting me to review the revised version of the manuscript PD-1 negative CD45RA effector-memory CD8 T-cells and CXCL10 macrophages underlie response to atezolizumab/bevacizumab in advanced hepatocellular carcinoma by Cappuyns et al. In this paper, authors study the role of PD1-negative CD8 T-cells and their role in response to Atezo/Bev. Through single-cell profiling, authors found that advanced HCC tumors with durable response to checkpoint blockade had an abundance of PDL1-expressing CXCL10+ macrophages and high levels of CXCL9/10/11, potentially attracting CXCR3+ effector-memory T-cells (CD8 TEM) into the tumor. These CD8 TEM cells preferentially differentiated into clonally-expanded, PD1-negative, CD45RA effector-memory CD8 T-cells (CD8 TEMRA) with strong cytotoxic activity, highlighting a unique mechanism of response to immunotherapy in advanced HCC.

Authors have addressed my concerns and expanded the number of samples. Some of the language has also been modified. However, I have a few suggestions to further clarify the statements in the manuscript. Would also suggest improvements to the figure legends.

We kindly thank Reviewer #1 for their time and critical review and have adapted the text as suggested (see below for details).

Title says "PD-1 negative CD45RA effector-memory CD8 T-cells and CXCL10 macrophages 1 underlie response to atezolizumab/bevacizumab in advanced hepatocellular carcinoma ". This still suggests a causal link between these immune subsets and response to Atezo/Bev. Would suggest change to "PD-1 negative CD45RA effector-memory CD8 T-cells and CXCL10 macrophages are associated with response to atezolizumab/bevacizumab in advanced hepatocellular carcinoma ".

We understand that the original title was too suggestive of a causal link. Therefore, we propose the following:

"PD-1 negative CD45RA effector-memory CD8 T-cells, CXCL10 macrophages and response to atezolizumab/bevacizumab in advanced hepatocellular carcinoma"

In the abstract they say "Tumours with durable response were enriched for PDL1- expressing CXCL10+ macrophages and, based on cell-cell interaction, expressed high levels of CXCL9/10/11 to attract peripheral CXCR3+ effector-memory T-cells (CD8 TEM) into the tumour." This suggests a functional link. Would stick to correlative language. "Tumours with durable response were enriched for PDL1- expressing CXCL10+ macrophages which also express high levels of CXCL9/10/11 and potentially recruit peripheral CXCR3+ effector-memory T-cells (CD8 TEM) into the tumour."

The abstract has been adapted as follows: "Tumours with durable response were enriched for PDL1-expressing CXCL10+ macrophages and, based on cell-cell interaction analysis, expressed high levels of CXCL9/10/11 to potentially attract peripheral CXCR3+ effector-memory T-cells (CD8 TEM) into the

tumour.”

Please clarify in figure legends, not clear from the figure legends-

Fig 2a-2d- UMAP of which samples ?

Fig. 2a and c refer to tumour biopsies (n=38). Fig. 2b and d refer to peripheral blood samples (n=72). Figure legends have been adapted to include number of samples.

Fig 2e and 2g says Atezo-Bev n=20. Figure 1 said this group was 25?. How many in this graph are responders vs.nonresponders.

Fig. 1 refers to all patients included in our study (n=44), of which 25 were treated with atezo/bev (15 responders versus 10 non-responders). As not all samples were available for all patients (please refer to **Supplementary Table 2** for details), the numbers in the subsequent figure legends may be different. More specifically, Fig. 2e and g include data from tumour biopsies only, of which 20 were treated with atezo/bev (12 responders vs 8 non-responders). Details were added to the figure legends.

Fig 2f How many T cells in responders and nonresponders?

Total of 4313 CD8 T-cells in atezo/bev treated patients: 3425 CD8 T-cells from 12 responders versus 888 CD8 T-cells from 8 non-responders. Numbers were added to the figure legend.

Fig 3a-b, how many responders and non-responders

Fig. 3a-b refers to patients treated with atezo/bev and included in the ‘TCR sharing’ analysis, comprising a total of 17 patients: 10 responders versus 7 non-responders.

Figure 4- how many T cells?

Total of 8989 intra-tumoural CD8 T-cells. Numbers were added to figure legend of Fig. 4a.

Figure 5b, how many samples, how many responders and non-responders

12 responders versus 8 non-responders. Numbers were added in the figure legend.

Fig 5c- why are there 20 patients to show PDL1 expression but only 17 for macro CXCL10

In 3 out of the 20 atezo/bev treated patients no CXCL10+ macrophages were detected within the tumour. For those patients, the PDL1 expression in Macro CXCL10 cannot be calculated.

Reviewer #2 (expert in hepatocellular carcinoma, biomarkers of drug resistance):

The authors have done a substantial amount of work in this revision, which is fantastic. I have only a minor concern about the use of "unique" in the last sentence of the abstract, as this study does not really offer a direct comparison of atezo bev versus other therapy outcomes.

We want to thank Reviewer #2 for their enthusiasm of our work. We removed “unique” from the last sentence of the abstract, as suggested.

Reviewer #3 (expert in computational biology, scRNAseq, and TCRseq):

The authors have almost addressed all my questions except that the gene expression matrix is maintained at the authors' own website instead of being deposited to a public database. Although it is not ideal, it is acceptable. Overall, it is a great work.

We thank Reviewer #3 for their time and critical review. We also want to emphasize that, within the limitations of the European GDPR law, we are committed to sharing our data with the scientific community, not only to ensure reproducibility of our findings, but also enable future research endeavours. Therefore, the raw sequencing reads of the scRNAseq, scTCRseq and Totalseq experiments have been deposited in the European Genome-phenome Archive (EGAS00001007547). Upon request, the raw data will be released via a Data Transfer Agreement after approval by the UZLeuven-VIB data access committee. Additionally, and to ensure reproducibility and accessibility, the read count data necessary to reproduce all analyses included in this manuscript will also be uploaded to our website for download.